# DMBP: DIFFUSION MODEL-BASED PREDICTOR FOR ROBUST OFFLINE REINFORCEMENT LEARNING AGAINST STATE OBSERVATION PERTURBATIONS

**Zhihe Yang**[1,2]    **Yunjian Xu**[1,2] [*]
[1]The Chinese University of Hong Kong, Hong Kong SAR, China
[2]The Chinese University of Hong Kong, Shenzhen Research Institute (SZRI), Guangdong, China
`zhyang@link.cuhk.edu.hk, yjxu@mae.cuhk.edu.hk`

## ABSTRACT

Offline reinforcement learning (RL), which aims to fully explore offline datasets for training without interaction with environments, has attracted growing recent attention. A major challenge for the real-world application of offline RL stems from the robustness against state observation perturbations, *e.g.*, as a result of sensor errors or adversarial attacks. Unlike online robust RL, agents cannot be adversarially trained in the offline setting. In this work, we propose Diffusion Model-Based Predictor (DMBP) in a new framework that recovers the actual states with conditional diffusion models for state-based RL tasks. To mitigate the error accumulation issue in model-based estimation resulting from the classical training of conventional diffusion models, we propose a non-Markovian training objective to minimize the sum entropy of denoised states in RL trajectory. Experiments on standard benchmark problems demonstrate that DMBP can significantly enhance the robustness of existing offline RL algorithms against different scales of random noises and adversarial attacks on state observations. Further, the proposed framework can effectively deal with incomplete state observations with random combinations of multiple unobserved dimensions in the test. Our implementation is available at `https://github.com/zhyang2226/DMBP`.

## 1 INTRODUCTION

Reinforcement learning (RL) has been proven to be a powerful tool for high-dimensional decision-making problems under uncertainty (Mnih et al., 2015; Silver et al., 2017; Schrittwieser et al., 2020). However, its trial-and-error learning manner requires frequent interactions with the environment, which can be expensive and/or dangerous in a variety of real-world applications (Levine et al., 2020). A widely adopted solution is to build up a simulator for policy training, which is costly and may fail due to the discrepancy between the simulator and reality. As a promising alternative that has received growing attention, offline RL fully explores offline datasets and requires no interaction with the environments in the training process.

A major challenge of offline training is on the robustness against perturbation on state observations, which may result from sensor errors, adversarial attacks, and mismatches between statistic datasets and the real environment. For example, GPS signal errors can lead to inaccurate positioning of autonomous vehicles, and position sensor errors can lead to erroneous estimation of robot arm postures. The robustness of the trained policy against state perturbations is vital for preventing agents from catastrophic movements. In online settings, various adversarial training methods have been proposed to robustly handle the mismatch between observed and actual states (Zhang et al., 2020; 2021; Sun et al., 2021). These methods are not directly applicable in offline training.

A classical approach against perturbed state observation is to train robust policies against worst-case disturbances (see the left subplot in Figure 1), which may lead to over-conservatism (Zhang et al., 2020; 2021). In a pioneering work (Yang et al., 2022), the authors propose an alternative approach that adopts the conservative smoothing method to smoothen the Q-value and regularize the policy, preventing the agent from taking catastrophic movements under adversarial attacks in the test. The performance of the aforementioned approach may decay quickly with the increasing noise scale, especially in complicated environments with high-dimensional action and state spaces.

---

[*]Corresponding author

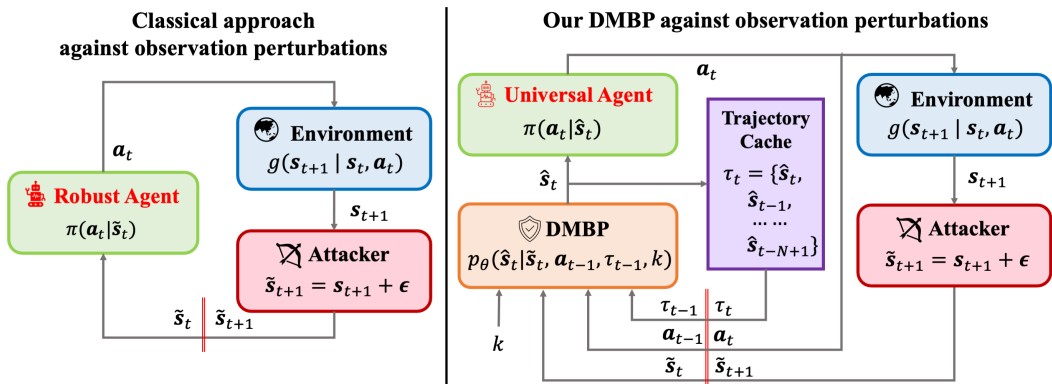

Figure 1: Overview diagram of classical approach against observation perturbations (left) and the proposed DMBP improved RL decision process against observation perturbations (right).

For online image-based deep RL, Lin et al. (2017) propose a model-based approach to "denoise" the observations by predicting the actual states. They construct a multiple-layer perceptron (MLP) neural network to detect the adversarial attack on image-based observations and predict the original states for decision-making in Atari games. For state-based RL tasks, similar MLP-based prediction methods have been used as data augmentation in online (Janner et al., 2019) and offline (Yu et al., 2020; 2021) settings *instead of* denoising tools. In general, MLP-based prediction methods cannot be applied to complicated state-based tasks (like Mujoco) which are sensitive to observation noise and prone to error accumulation.

Recently, diffusion-based generative models (Sohl-Dickstein et al., 2015; Ho et al., 2020; Song et al., 2020b) are widely used in offline RL/decision-making problems as trajectory generators (Janner et al., 2022; Ajay et al., 2022; Liang et al., 2023) and behavior cloners (Wang et al., 2022). We note that the potential of diffusion models to facilitate decision making via state denoising has not been fully explored.

Towards this end, we propose a new framework that predicts the actual states against observation perturbations for state based offline RL, which is referred to as *Diffusion Model-Based Predictor (DMBP)*. Different from the aforementioned works, the proposed approach utilizes diffusion models as **noise reduction** tools rather than generation models, and can therefore enhance the robustness of existing offline RL algorithms against different scales of perturbations on state observations.

A diagram of the proposed approach is shown in the right subplot of Figure 1. Given the past-estimated state trajectory, last-step agent-generated action, and the noised current state from the environment, DMBP utilizes a conditioned diffusion model to estimate the current state by reversely denoising data. To mitigate the error accumulation issue in state estimation, we propose a new non-Markovian loss function that minimizes the sum entropy of denoised states over the RL trajectory (cf. Section 4). In order to well capture the relationship between the noised current state and the denoised state trajectory (especially the last RL timestep denoised state), we propose an Unet-MLP neural network structure to predict noise information (cf. Appendix B.1). The output of DMBP is an estimation of the current state, which is fed into an offline RL algorithm to generate the action. To our knowledge, this is the *first* state denoising framework for offline RL against observation perturbations in state-based tasks.

The proposed framework has several advantages over existing offline RL methods against noisy observations. First, with an objective of recovering the actual state, DMBP can significantly strengthen the robustness of existing offline RL algorithms against different scales of random noises and adversarial attacks. The proposed approach does not lead to over-conservative policies, compared with counterparts that train robust policies against worst-case (or adversarial) perturbations.

Further, by virtue of the capability of diffusion models to infill the missing regions (*i.e.*, image inpainting), DMBP facilitates the decision making under incomplete state observations with random combinations of multiple unobserved dimensions in the test. Such a situation is common in reality, for example, when robots continue to work with compromised sensors.

## 2    RELATED WORKS

**Robust RL.**    Robust RL can be categorized into two taxonomies: training-time and testing-time robustness.    Training-time robust RL involves perturbations during the training process, while evaluating the agent in a clean environment (Zhang et al., 2022b; Ye et al., 2023).    Conversely, testing-time robust RL focuses on training the agent with unperturbed datasets or environments and then testing its performance in the presence of disturbances (Yang et al., 2022; Panaganti et al., 2022). Our work primarily aims at enhancing the testing-time robustness of existing offline RL algorithms.

Testing-time robust RL formulations can generally be divided into three categories (Xu et al., 2022). i) *Uncertain observations*: In online settings, Zhang et al. (2020) propose a state-adversarial Markov decision process (SA-MDP) framework, which is advanded by Zhang et al. (2021); Sun et al. (2021) that adopt neural networks to simulate worst-case observation attacks for the training of more robust policies.    In offline settings, Yang et al. (2022) utilize the conservative smoothing method to make the agent take similar actions when the perturbations on state observation are relatively small. ii) *Uncertain actions*: Tessler et al. (2019) explore the training of robust policies against two types of action uncertainties, *i.e.*, occasional and constant adversarial perturbations. Tan et al. (2020) utilize adversarial training on actions to enhance the robustness against action perturbations. iii) *Uncertain transitions and rewards*: The computation of optimal policies against uncertain environment parameters has been explored under the robust Markov Decision Process (MDP) (Xu & Mannor, 2006; Roy et al., 2017; Ho et al., 2018) and the distributionally robust MDP frameworks (Xu & Mannor, 2010; Yu & Xu, 2015). In online RL settings, Pinto et al. (2017) and Gleave et al. (2019) train the agent under adversarial model uncertainty through a two-player Markov game approach (Littman, 1994). For offline RL training, Panaganti et al. (2022) propose a dual reformulated robust Bellman operator to deal with the uncertain transition probability.

For models in the first two categories, the true states and transition probabilities of the environments are not influenced by the action, which is not the case for the robust approaches against model uncertainties developed in the third category. Our work belongs to the first category.

**Diffusion models in offline RL.**    The diffusion model was originally proposed as an iterative denoising procedure for image generation in *computer vision* (Sohl-Dickstein et al., 2015; Ho et al., 2020). Recently, diffusion model has been adopted in decision-making for state-based tasks. *Diffuser* (Janner et al., 2022) and *Decision Diffuser* (Ajay et al., 2022) utilize the conditional diffusion model as a trajectory generator to facilitate the decision making of the agent. Wang et al. (2022) propose the *Diffusion-QL* algorithm that adopts the diffusion model to regulate the policy not to be far away from the one used in datasets, in a similar spirit to Fujimoto et al. (2019); Wu et al. (2019). Different from the aforementioned works, the proposed approach utilizes the diffusion model as a denoiser (against state observation perturbations) rather than a generator, for robust offline training of RL agents.

## 3    PRELIMINARIES

**Offline RL.**    RL tasks are generally modeled as Markovian Decision Processes (MDP) in the form of $M = (\mathcal{S}, \mathcal{A}, r, P, \gamma, d_0)$, where $\mathcal{S}$ is the state space, $\mathcal{A}$ is the action space, $r : \mathcal{S} \times \mathcal{A} \to \mathbb{R}$ represents the reward function, $P$ is the model dynamics, $\gamma \in [0, 1)$ is the discount factor, and $d_0 \in \Delta(\mathcal{S})$ is the distribution of initial state $\boldsymbol{s}_0$ (the set of the probability distribution over $\mathcal{X}$ is denoted as $\Delta(\mathcal{X})$). $P(\boldsymbol{s}'|\boldsymbol{s}, \boldsymbol{a}) : \mathcal{S} \times \mathcal{A} \to \Delta(\mathcal{S})$ represents the transition function from state $\boldsymbol{s}$ to $\boldsymbol{s}'$ when taking action $\boldsymbol{a}$. The state-action-reward transitions over trajectory are recorded as $\boldsymbol{\tau} := (\boldsymbol{s}_t, \boldsymbol{a}_t, r_t)_{t \geq 0}$.The goal of RL is to learn a policy $\pi_\phi$ that maximizes the expectation of the cumulated discounted reward $R(\boldsymbol{\tau}) = \sum_{t=0}^{\infty} \gamma^t r(\boldsymbol{s}_t, \boldsymbol{a}_t)$, denoted by $\pi_\phi^* = \arg\max_\pi \mathbb{E}_{\boldsymbol{s}_0 \sim \boldsymbol{d}_0, \boldsymbol{a} \sim \pi}[R(\boldsymbol{\tau})]$.

A commonly used iteration method for state-based tasks is under the *actor-critic* framework, where the Q-value of a policy is defined as $Q_\pi(\boldsymbol{s}_t, \boldsymbol{a}_t) := \mathbb{E}_{\boldsymbol{a} \sim \pi}[\sum_{i=t}^{\infty} \gamma^{(i-t)} r(\boldsymbol{s}_i, \boldsymbol{a}_i)]$ and is modeled using neural networks (recorded as $Q_\psi(\boldsymbol{s}_t, \boldsymbol{a}_t)$). To approach an optimal policy, the temporal difference (TD) method is adopted to update the critic Q-value by minimizing the TD loss: $\mathcal{L}_{TD}(\psi) := \mathbb{E}_{(\boldsymbol{s}, \boldsymbol{a}, r, \boldsymbol{s}') \in \mathcal{D}}[(r + \gamma \max_{\boldsymbol{a}' \in \mathcal{A}} Q_\psi(\boldsymbol{s}', \boldsymbol{a}') - Q_\psi(\boldsymbol{s}, \boldsymbol{a}))^2]$. The actor is updated by $\mathcal{L}_{actor}(\phi) := \mathbb{E}_{\boldsymbol{s} \in \mathcal{D}, \boldsymbol{a} \sim \pi_\phi(\cdot|\boldsymbol{s})}[-Q(\boldsymbol{s}, \boldsymbol{a})]$, where the dataset $\mathcal{D}$ records historical interactions between agent and environment, and is continuously updated in the alternate training of the actor and the critic. In offline RL settings, the training is performed on a statistic dataset $\mathcal{D}_\nu := \{(\boldsymbol{s}, \boldsymbol{a}, r, \boldsymbol{s}')\}$, which is obtained from a behavior policy $\pi_\nu$ without any interaction with the environment.

Direct adoption of the actor-critic approach may lead to a severe distributional shift between the trained policy $\pi_\phi$ and the behavior policy $\pi_\nu$ due to the over-estimation of the Q-value of actions unseen in datasets. To mitigate this issue, *policy regularization* has been adopted to update the actor through constrained policy loss (Wu et al., 2019; Kumar et al., 2019; Fujimoto et al., 2019; Fujimoto & Gu, 2021; Wang et al., 2022): $\mathcal{L}(\phi) := \mathcal{L}_d(\phi) + \alpha_{actor}\mathcal{L}_{actor}(\phi)$, where $\mathcal{L}_d(\phi)$ is the behavior cloning loss representing the nominal distance between the trained policy and the behavior policy, and $\alpha_{actor}$ is the coefficient for the Q-value term. Alternatively, *conservative Q estimation* updates the critic through minimizing the constrained Q-value loss (Kumar et al., 2020; An et al., 2021; Lyu et al., 2022; Yang et al., 2022): $\mathcal{L}(\psi) := \mathcal{L}_q(\psi) + \alpha_{critic}\mathcal{L}_{TD}(\psi)$, where $\mathcal{L}_q(\psi)$ is the penalty on Q-value for out-of-distribution actions, and $\alpha_{critic}$ is the coefficient for the TD loss term.

**Diffusion model.**   Diffusion based generative models have been widely used for synthesizing high-quality images from text descriptions. The *forward process*, *i.e.*, the noising process, is a Markov chain that gradually adds Gaussian noise to data according to a variance schedule $\beta_1, \ldots, \beta_K$:

$$q(\boldsymbol{x}_{1:K} \mid \boldsymbol{x}_0) := \prod_{k=1}^{K} q(\boldsymbol{x}_k \mid \boldsymbol{x}_{k-1}), \quad q(\boldsymbol{x}_k \mid \boldsymbol{x}_{k-1}) := \mathcal{N}(\boldsymbol{x}_k; \sqrt{1-\beta_k}\boldsymbol{x}_{k-1}, \beta_k\boldsymbol{I}).$$

The *reverse process*, *i.e.*, the denoising process, is a Markov chain with learned Gaussian transitions that usually starts at $p(x_K) = \mathcal{N}(x_K; 0, \boldsymbol{I})$:

$$p_\theta(\boldsymbol{x}_{0:K}) := p(\boldsymbol{x}_K) \prod_{k=1}^{K} p_\theta(\boldsymbol{x}_{k-1} \mid \boldsymbol{x}_k), \quad p_\theta(\boldsymbol{x}_{k-1} \mid \boldsymbol{x}_k) := \mathcal{N}(\boldsymbol{x}_{k-1}; \boldsymbol{\mu}_\theta(\boldsymbol{x}_k, k), \boldsymbol{\Sigma}_\theta(\boldsymbol{x}_k, k)).$$

Ho et al. (2020) derive a simplified surrogate loss for the *reverse process* denoising:

$$\mathcal{L}_{\text{denoise}}(\theta) := \mathbb{E}_{k\sim[1,K], \boldsymbol{\epsilon}\sim\mathcal{N}(\boldsymbol{0},\boldsymbol{I})}[\|\boldsymbol{\epsilon}_\theta(\boldsymbol{x}_k, k) - \boldsymbol{\epsilon}\|^2]. \tag{1}$$

The Gaussian noise $\boldsymbol{\epsilon}$, which perturbs the original data $\boldsymbol{x}_0$ to noised data $\boldsymbol{x}_k$, is estimated through the neural network based predictor $\boldsymbol{\epsilon}_\theta(\boldsymbol{x}_k, k)$. $\boldsymbol{x}_{k-1}$ is sampled from the reverse process as $\boldsymbol{\mu}_\theta(\boldsymbol{x}_k, k)$ and $\boldsymbol{\Sigma}_\theta(\boldsymbol{x}_k, k)$ are functions of $\boldsymbol{\epsilon}_\theta(\boldsymbol{x}_k, k)$. It is straightforward to extend diffusion models to conditional ones with $p_\theta(\boldsymbol{x}_{t-1} \mid \boldsymbol{x}_t, c)$ (conditioned on information $c$), where the noise prediction is given by $\boldsymbol{\epsilon}_\theta(\boldsymbol{x}_k, k, c)$.

## 4   DIFFUSION MODEL BASED PREDICTOR

We express the perturbed version of the original state $\boldsymbol{s}$ as $\tilde{\boldsymbol{s}}$, where $\mathbb{B}_d(\boldsymbol{s}, \boldsymbol{\epsilon}) := \{\tilde{\boldsymbol{s}} : d(\boldsymbol{s}, \tilde{\boldsymbol{s}}) \le \boldsymbol{\epsilon}\}$ is the perturbation set and the metric $d(\cdot, \cdot)$ is based on $\ell_p$ norm, as in Shen et al. (2020). An adversarial attack on state $\boldsymbol{s}$ is introduced in Yang et al. (2022): $\tilde{\boldsymbol{s}}^* = \arg\max_{\tilde{\boldsymbol{s}}\in\mathbb{B}_d(\boldsymbol{s},\boldsymbol{\epsilon})} D(\pi_\phi(\cdot|\boldsymbol{s})\|\pi_\phi(\cdot|\tilde{\boldsymbol{s}}))$, where $D(\cdot\|\cdot)$ is the divergence of two distributions. The targets of both works are to minimize the smoothness regularizer for the policy: $\mathcal{R}_s^\pi = \mathbb{E}_{\boldsymbol{s}\in\mathcal{D}} \max_{\tilde{\boldsymbol{s}}\in\mathbb{B}_d(\boldsymbol{s},\boldsymbol{\epsilon})} D(\pi(\cdot|\boldsymbol{s})\|\pi(\cdot|\tilde{\boldsymbol{s}}))$, and to minimize the smoothness regularizer for the value function: $\mathcal{R}_s^V = \mathbb{E}_{\boldsymbol{s}\in\mathcal{D}, \boldsymbol{a}\sim\pi} \max_{\tilde{\boldsymbol{s}}\in\mathbb{B}_d(\boldsymbol{s},\boldsymbol{\epsilon})}(Q(\boldsymbol{s},\boldsymbol{a}) - Q(\tilde{\boldsymbol{s}},\boldsymbol{a}))$, against the perturbations on state observations. We remark that we do not normalize the state observations when applying the perturbations as in Shen et al. (2020); Sun et al. (2021), in contrast to Zhang et al. (2020); Yang et al. (2022).

In Section 4.1, we propose DMBP to recover the actual state for decision-making (which is fundamentally different from the technical approaches in aforementioned works). In Section 4.2, we propose a new non-Markovian loss function to mitigate error accumulation. In Section 4.3, we apply DMBP to RL tasks under incomplete state observations with unobserved dimension(s).

### 4.1   CONDITIONAL DIFFUSION FOR PREDICTING REAL STATE

As there are two timesteps involved in our framework, we use superscripts $i, k \in \{1, \ldots K\}$ to denote diffusion timestep and subscript $t \in \{1, \ldots, T\}$ to denote trajectory timestep in RL tasks.

DMBP is inspired by the diffusion model framework originally proposed for image generations (Ho et al., 2020). As the proposed framework essentially deals with information with small to medium scale noises instead of generating data from pure noise, we redesign the variance schedule as:

$$\beta_i = 1 - \alpha_i = e^{-\frac{b}{i+a}+c}, \quad \bar{\alpha}_k = \prod_{i=1}^{k} \alpha_i, \quad \tilde{\beta}_i = \frac{1-\bar{\alpha}_{i-1}}{1-\bar{\alpha}_i}\beta_i,$$

where $a, b, c$ are hyperparameters (cf. Appendix B.2). The redesigned variance schedule restricts the noise scale to be small in the diffusion process and limits the total number of diffusion timesteps $K$ for predictor training. We use the conditional diffusion model to obtain the denoised state $\hat{s}_t$ from the noised state $\tilde{s}_t$, with the condition on last step action $a_{t-1}$ and the previously denoised state trajectory $\tau_{t-1}^{\hat{s}} := \{\hat{s}_1, \hat{s}_2, ..., \hat{s}_{t-1}\}$. The denoised state $\hat{s}_t$ is sampled from the reverse denoising process, which can be expressed as a Markov chain:

$$\hat{s}_t \sim p_\theta(\tilde{s}_t^{0:k} \mid a_{t-1}, \tau_{t-1}^{\hat{s}}) = f_k(\tilde{s}_t) \prod_{i=1}^{k} p_\theta(\tilde{s}_t^{i-1} \mid \tilde{s}_t^i, a_{t-1}, \tau_{t-1}^{\hat{s}}), \tag{2}$$

where $f_k(\tilde{s}_t) = \sqrt{\bar{\alpha}_k}\tilde{s}_t$. The transitions $p_\theta(\tilde{s}_t^{i-1} \mid \tilde{s}_t^i, a_{t-1}, \tau_{t-1}^{\hat{s}})$ can be modeled using Gaussian distribution $\mathcal{N}(\tilde{s}_t^{i-1}; \mu_\theta(\tilde{s}_t^i, a_{t-1}, \tau_{t-1}^{\hat{s}}, i), \Sigma_\theta(\tilde{s}_t^i, a_{t-1}, \tau_{t-1}^{\hat{s}}, i))$, with the following mean and variance (Ho et al., 2020):

$$\mu_\theta(\tilde{s}_t^i, a_{t-1}, \tau_{t-1}^{\hat{s}}, i) = \frac{\sqrt{\alpha_i}(1-\bar{\alpha}_{i-1})}{1-\bar{\alpha}_i}\tilde{s}_t^i + \frac{\sqrt{\bar{\alpha}_{i-1}}\beta_i}{1-\bar{\alpha}_i}\tilde{s}_t^{0(i)}, \quad \Sigma_\theta(\tilde{s}_t^i, a_{t-1}, \tau_{t-1}^{\hat{s}}, i) = \tilde{\beta}_i I.$$

Here, $\tilde{s}_t^{0(i)}$ is the state directly recovered from the current diffusion step noise prediction, which is given by

$$\tilde{s}_t^{0(i)} = \frac{1}{\sqrt{\bar{\alpha}_i}}[\tilde{s}_t^i - \sqrt{1-\bar{\alpha}_i}\epsilon_\theta(\tilde{s}_t^i, a_{t-1}, \tau_{t-1}^{\hat{s}}, i)]. \tag{3}$$

The reverse diffusion chain is given by

$$\tilde{s}_t^{i-1} \mid \tilde{s}_t^i = \frac{\tilde{s}_t^i}{\sqrt{\alpha_i}} - \frac{\beta_i}{\sqrt{\alpha_i(1-\bar{\alpha}_i)}}\epsilon_\theta(\tilde{s}_t^i, a_{t-1}, \tau_{t-1}^{\hat{s}}, i) + \sqrt{\tilde{\beta}_i}\epsilon, \tag{4}$$

where $\epsilon \sim \mathcal{N}(0, I)$ and is set to be $0$ at the final denoising step ($i = 1$). For the final step denoising output of Eq. 4 $\tilde{s}_t^0$ (i.e., the output of DMBP), we refer it to as $\hat{s}_t$. $\hat{s}_t$ can be used for decision-making by any offline-trained agent according to $a_t = \pi_\phi(\cdot \mid \hat{s}_t)$. $\hat{s}_t$ is stored in the trajectory cache $\tau_t^{\hat{s}}$, and the pair of $(\tau_t^{\hat{s}}, a_t)$ will be utilized for the next step denoising. In practice, on account of the stochasticity involved in the diffusion process, we denoise the state 50 times in parallel and take the average value as the final output $\hat{s}_t$ to prevent the denoised state from falling out of the distribution.

We find that directly inputting state trajectories and action into neural networks leads to poor noise estimation (cf. Appendix C for ablation study on network structure), partially due to the fact that $\hat{s}_{t-1}$ is more closely related to $\tilde{s}_t^i$ than $\hat{s}_j$ with $j < t-1$, and that this information cannot be well captured by neural networks. Therefore, we first extract the information from the trajectory with U-net (Ronneberger et al., 2015; Janner et al., 2022) (recorded as $U_\xi(\tilde{s}_t^i, \tau_{t-1}^{\hat{s}})$), and then utilize an MLP-based neural network to predict the noise through $\epsilon_\theta(U_\xi(\tilde{s}_t^i, \tau_{t-1}^{\hat{s}}), \tilde{s}_t^i, a_{t-1}, \hat{s}_{t-1}, i)$, which is represented by $\epsilon_\theta(\tilde{s}_t^i, a_{t-1}, \tau_{t-1}^{\hat{s}}, i)$ for notational convenience. See Appendix B.1 for details.

## 4.2 NON-MARKOVIAN LOSS FUNCTION

The accuracy of the current denoising result $\hat{s}_t$ is highly dependent on the accuracy of the diffusion condition $\tau_{t-1}^{\hat{s}}$. A straightforward adoption of the loss function 1 in the denoising diffusion probabilistic model (DDPM) (Ho et al., 2020) may lead to severe error accumulation in online testing, due to the mismatch between the training-process noise prediction $\epsilon_\theta(\tilde{s}_t^i, a_{t-1}, \tau_{t-1}^s, i)$ and the testing-process noise prediction $\epsilon_\theta(\tilde{s}_{t+1}^i, a_t, \tau_{t-1}^{\hat{s}}, i)$. To mitigate error accumulation and enhance the robustness of DMBP, we propose a non-Markovian training objective to minimize the sum entropy of denoised states over the RL trajectory $\tau$:

$$\mathcal{L}_{\text{entropy}} = \sum_{t=2}^{T} \mathbb{E}_{s_t \in \tau, q(s_t)} \left[ -\log P(\hat{s}_t \mid a_{t-1}, \tau_{t-1}^{\hat{s}}) \right], \tag{5}$$

where $P(\hat{s}_t \mid a_{t-1}, \tau_{t-1}^{\hat{s}}) = p_\theta(\tilde{s}_t^0 \mid a_{t-1}, \tau_{t-1}^{\hat{s}})$ is the distribution of state after denoising at RL timestep $t$. Following the setting in (Ho et al., 2020), we establish a closed-form expression of the training objective that minimizes Eq. 5, which can be simplified as (cf. the details in Appendix A):

$$\mathcal{L}_{\text{simple}}(\theta) = \mathbb{E}_{s_1 \sim d_0, \epsilon_t^i \sim \mathcal{N}(0,I), i \sim \mathcal{U}_K} \left[ \sum_{t=2}^{T} \|\epsilon_\theta(\tilde{s}_t^i, a_{t-1}, \tau_{t-1}^{\hat{s}}, i) - \epsilon_t^i\|^2 \right], \tag{6}$$

where $\mathcal{U}_K$ is the uniform distribution over discrete set $\{1, 2, \ldots, K\}$, and the noised states for all terms are sampled through:

$$\tilde{s}_t^i = \sqrt{\bar{\alpha}_i} s_t + \sqrt{1 - \bar{\alpha}_i} \epsilon_t^i, \quad \epsilon_t^i \sim \mathcal{N}(\mathbf{0}, \mathbf{I}).$$

For computational convenience, we further simplify Eq. 6 and derive our non-Markovian loss function by sampling the partial trajectory $(s_{t-N}, a_{t-N}, s_{t-N+1}, \ldots, s_{t+M-1})$ from the offline dataset $\mathcal{D}_\nu$ ($N$ is the condition trajectory length and $M$ is the sample trajectory length):

$$\mathcal{L}(\theta) = \mathbb{E}_{i \sim \mathcal{U}_K, \epsilon_t \sim \mathcal{N}(\mathbf{0}, \mathbf{I}), (s_{t-N}, \ldots, s_{t+M-1}) \in \mathcal{D}_\nu} \left[ \underbrace{\|\epsilon_\theta(\tilde{s}_t^i, a_{t-1}, \tau_{t-1}^s, i) - \epsilon_t^i\|^2}_{L_t} + \right.$$
$$\left. \sum_{m=t+1}^{t+M-1} \underbrace{\|\epsilon_\theta(\tilde{s}_m^i, a_{m-1}, \tau_{m-1}^{\breve{s}}, i) - \epsilon_m^i\|^2}_{L_m} \right], \tag{7}$$

where the state trajectory condition for the predictor $\epsilon_\theta$ in $L_t$ is the original $\tau_{t-1}^s = \{s_{t-N}, \ldots, s_{t-1}\}$ from the offline dataset $\mathcal{D}_\nu$, and the state trajectory condition in $L_m$ can be expressed as $\tau_{m-1}^{\breve{s}} = \{\breve{s}_j \mid j \in \{m-N, \ldots, m-1\}\}$, with:

$$\breve{s}_j = \begin{cases} s_j & \text{if } j < t, \\ \frac{1}{\sqrt{\bar{\alpha}_i}} \left[ \tilde{s}_j^i - \sqrt{1 - \bar{\alpha}_i} \epsilon_\theta(\tilde{s}_j^i, a_{j-1}, \tau_{j-1}^{\breve{s}}, i) \right] & \text{otherwise } (j \in \{t, \ldots, t+M-2\}). \end{cases}$$

Different from the loss function in Ho et al. (2020) that concerns only single-step diffusion accuracy (for data generation under conditions on ground-truth states), the proposed non-Markovian loss function trades off between the diffusion accuracy at the current RL timestep and the condition shift in a long RL time horizon (to avoid error accumulation).

## 4.3 DIFFUSION BASED STATE INFILLING FOR UNOBSERVED DIMENSION

Inspired by the application of diffusion models in image inpainting (Lugmayr et al., 2022), we propose a state infilling procedure for DMBP facilitated decision making, which is shown to work well on state-based RL tasks with incomplete state observations (cf. Section 5.2).

We denote the ground truth state as $s_t$, the unobserved state information as $(1 - m) \odot s_t$, and the observed state information as $\acute{s}_t = m \odot s_t$, which is incomplete with some masked dimensions. Given the recovered state trajectory $\tau_{t-1}^{\hat{s}}$, agent generated action $a_{t-1}$, and the known information of the current state $\acute{s}_t$, DMBP aims to recover the original state $\hat{s}_t$ for decision making. Following the inpainting method of Lugmayr et al. (2022), DMBP infills the missing state information through Algorithm 1. For each diffusion timestep, the known region of the state is determined from the forward process (noising process) in line

**Algorithm 1** Diffusion based state infilling for DMBP

**Require:** $s_t^K \sim \mathcal{N}(\mathbf{0}, \mathbf{I}), \acute{s}_t, a_{t-1}, \tau_{t-1}^{\hat{s}}, m$
1: **for** $i = K, \ldots, 1$ **do**
2:     **for** $u = 1, \ldots, U$ **do**
3:         $\epsilon \sim \mathcal{N}(\mathbf{0}, \mathbf{I})$ if $i > 1$, else $\epsilon = \mathbf{0}$
4:         $\acute{s}_{t,\text{known}}^{i-1} = \sqrt{\bar{\alpha}_i} \acute{s}_t + \sqrt{1 - \bar{\alpha}_i} \epsilon$
5:         $z \sim \mathcal{N}(\mathbf{0}, \mathbf{I})$ if $i > 1$, else $z = \mathbf{0}$
6:         $\epsilon_{\text{pred}} = \epsilon_\theta(\acute{s}_t^i, a_{t-1}, \tau_{t-1}^{\hat{s}}, i)$
7:         $\acute{s}_{t,\text{unknown}}^{i-1} = \frac{1}{\sqrt{\alpha_i}} \acute{s}_t^i - \frac{\beta_i}{\sqrt{\alpha_i(1-\bar{\alpha}_i)}} \epsilon_{\text{pred}} + \sqrt{\beta_i} z$
8:         $\acute{s}_t^{i-1} = m \odot \acute{s}_{t,\text{known}}^{i-1} + (1 - m) \odot \acute{s}_{t,\text{unknown}}^{i-1}$
9:         **if** $u < U$ and $i > 1$ **then**
10:           $\acute{s}_t^i \sim \mathcal{N}(\sqrt{1 - \beta_{i-1}} \acute{s}_t^{i-1}, \beta_{i-1} \mathbf{I})$
11:         **end if**
12:     **end for**
13: **end for**
14: **return** $\hat{s}_t = \acute{s}_t^0$

4, and the unknown region of the state is determined from the reverse process (denoising process) in line 7. To avoid the disharmony of the forward and reverse process generated information, the combined information (in line 8) takes one diffusion forward step in line 10, which is called "resampling". The resampling is performed by $U$ times for one diffusion timestep. More resampling times may lead to more accurate and harmonious diffusion information at the cost of higher computational load.

## 5 EXPERIMENTS

We evaluate the proposed DMBP together with several state-of-the-art baseline offline RL algorithms on D4RL Gym benchmark (Fu et al., 2020) against different types of attacks on state observations. The baseline algorithms include Batch Constrained deep Q-learning (*BCQ*) (Fujimoto et al., 2019), Conservative Q-Learning (*CQL*) (Kumar et al., 2020), TD3 with Behavior Cloning (*TD3+BC*) (Fujimoto & Gu, 2021), Diffusion Q-Learning (*Diffusion QL*) (Wang et al., 2022), and Robust Offline Reinforcement Learning (*RORL*) (Yang et al., 2022).

We train DMBP for 300 epochs (1000 gradient steps with a batch size of 256 for each epoch) with hyperparameters defined in Appendix B.2. We train the baseline algorithms with the corresponding suggested hyperparameters in specific environments and datasets. We perform two tests, on robustness against noised state observations (in Section 5.1) and on robustness against incomplete state observations with unobserved dimension(s) (in Section 5.2). We utilize the same DMBP for all baseline algorithms (cf. the framework in Figure 1), and benchmark their performance against the original baseline algorithms (without DMBP). We present partial results on the D4RL Mujoco benchmark, and the results of other datasets (including medium-expert, medium, and full-replay) and other benchmarks (including Adroit and Franka Kitchen) can be found in Appendix D.

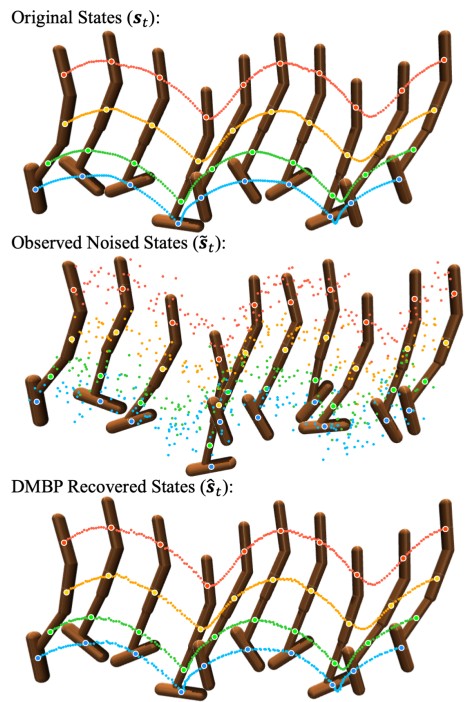

Figure 2: Visualization of the denoising effect of DMBP with Diffusion QL (trained on the dataset of hopper-meidum-replay-v2). The observation is perturbed with Gaussian distributed random noise with std of 0.10.

## 5.1 ROBUSTNESS AGAINST NOISED STATE OBSERVATIONS

Firstly, we evaluate the performance of DMBP in a basic setting with Gaussian noises of standard deviation $\kappa$: $\tilde{s}_t = s_t + \kappa \cdot \mathcal{N}(\mathbf{0}, \mathbf{I})$. The evaluation results in Table 1 indicate that DMBP can significantly enhance the robustness of all baseline algorithms, especially on the dataset "medium-replay", where DMBP strengthened baseline algorithms achieve similar scores as in the corresponding noise-free cases. To demonstrate the powerful denoising effect of DMBP, we visualize a partial trajectory of "hopper" during the test in Figure 2.

Table 1: D4RL score of the baseline algorithms ("base" recorded in black) and the DMBP strengthened ones ("DMBP" recorded in blue) trained with expert (e) and medium-replay (m-r) datasets under different scales of Gaussian random noises on state observation. The evaluation results are averaged over 5 random checkpoints (20 tests for each checkpoint).

| Env | Dataset | Noise scale | BCQ base | BCQ DMBP | CQL base | CQL DMBP | TD3+BC base | TD3+BC DMBP | Diffusion QL base | Diffusion QL DMBP | RORL base | RORL DMBP |
|---|---|---|---|---|---|---|---|---|---|---|---|---|
| HalfCheetah | e | 0 | 96.9±1.8 | - | 93.0±6.1 | - | 95.8±8.9 | - | 92.9±10.7 | - | 108.5±11.2 | - |
| | | 0.05 | 4.5±2.6 | 60.2±23.9 | 18.1±8.6 | 60.9±22.5 | 7.3±6.6 | 77.1±15.5 | 4.8±3.6 | 75.2±20.7 | 15.4±3.9 | 55.7±29.2 |
| | | 0.10 | 4.5±2.5 | 26.8±16.2 | 7.4±4.0 | 40.5±16.6 | 4.7±3.6 | 47.5±22.2 | 3.3±2.5 | 39.8±21.8 | 3.7±1.9 | 32.8±20.4 |
| | m-r | 0 | 41.6±4.2 | - | 47.0±1.0 | - | 45.2±0.9 | - | 47.7±0.8 | - | 66.7±1.4 | - |
| | | 0.10 | 20.6±6.9 | 38.5±11.2 | 35.6±1.3 | 45.8±1.0 | 28.5±5.5 | 44.3±1.0 | 30.1±4.1 | 45.6±0.9 | 43.5±2.4 | 61.9±1.2 |
| | | 0.15 | 14.8±10.2 | 35.1±8.7 | 28.8±1.5 | 44.6±1.1 | 24.0±8.9 | 42.5±2.6 | 24.2±7.6 | 44.6±3.0 | 30.3±5.9 | 58.4±1.2 |
| Hopper | e | 0 | 88.4±22.4 | - | 109.1±13.7 | - | 108.9±10.5 | - | 104.9±15.1 | - | 110.4±3.1 | - |
| | | 0.05 | 34.3±13.4 | 61.0±25.2 | 41.2±21.8 | 85.7±26.2 | 32.2±18.4 | 79.1±28.2 | 38.2±12.4 | 84.8±27.4 | 56.9±34.9 | 64.3±19.8 |
| | | 0.10 | 24.3±10.9 | 37.1±18.5 | 24.3±11.8 | 48.8±20.4 | 22.7±11.6 | 32.6±18.7 | 24.0±9.3 | 56.1±17.3 | 24.1±20.2 | 37.5±10.5 |
| | m-r | 0 | 78.7±19.6 | - | 96.9±8.8 | - | 80.9±24.5 | - | 95.7±17.2 | - | 103.1±0.8 | - |
| | | 0.10 | 15.7±9.0 | 66.8±17.3 | 47.5±21.6 | 89.1±12.4 | 14.4±12.3 | 71.9±24.5 | 25.9±12.4 | 85.9±20.9 | 85.9±29.5 | 103.2±1.3 |
| | | 0.15 | 11.1±7.2 | 64.5±17.2 | 33.7±21.2 | 80.7±16.5 | 9.6±7.3 | 66.1±22.8 | 17.9±11.5 | 72.2±22.9 | 51.1±22.3 | 104.2±3.2 |
| Walker2d | e | 0 | 111.6±0.6 | - | 108.8±1.9 | - | 110.7±0.5 | - | 109.6±0.5 | - | 104.8±12.5 | - |
| | | 0.10 | 77.9±37.6 | 110.3±2.0 | 97.6±21.9 | 94.3±20.3 | 72.9±39.4 | 109.2±1.5 | 93.3±27.2 | 109.1±4.0 | 95.4±19.7 | 97.8±20.2 |
| | | 0.15 | 28.2±32.4 | 104.2±13.5 | 78.9±33.2 | 83.4±23.3 | 9.2±13.6 | 107.5±5.2 | 30.5±32.5 | 94.5±18.1 | 81.6±26.4 | 84.5±26.4 |
| | m-r | 0 | 50.6±31.6 | - | 79.9±4.8 | - | 84.7±9.8 | - | 93.1±10.9 | - | 88.7±1.9 | - |
| | | 0.10 | 14.7±11.1 | 53.1±28.5 | 70.8±18.9 | 78.7±7.2 | 40.7±25.3 | 84.4±8.7 | 59.6±31.8 | 92.6±10.6 | 88.6±1.1 | 88.4±2.5 |
| | | 0.15 | 11.2±5.9 | 52.9±29.9 | 48.6±26.5 | 73.6±10.1 | 16.5±12.8 | 77.9±17.2 | 19.2±15.7 | 91.3±9.6 | 89.4±1.2 | 89.0±4.5 |

Table 2: D4RL score of the baseline algorithms and the DMBP strengthened ones under uniformly distributed random noise (U-rand), maximum action-difference attack (MAD), and minimum Q-value attack (MinQ) on state observations.

| Env | Dataset/ Noise Scale | Noise Type | BCQ base | BCQ DMBP | CQL base | CQL DMBP | TD3+BC base | TD3+BC DMBP | Diffusion QL base | Diffusion QL DMBP | RORL base | RORL DMBP |
|---|---|---|---|---|---|---|---|---|---|---|---|---|
| HalfCheetah | e 0.05 | U-rand | 7.4±4.9 | 69.1±21.5 | 27.2±6.4 | 69.6±22.4 | 16.3±13.1 | 84.2±17.1 | 11.6±10.9 | 77.8±21.8 | 24.3±7.5 | 66.8±27.0 |
| | | MAD | 3.6±1.7 | 52.5±17.9 | 12.4±6.9 | 61.2±19.7 | 4.7±3.5 | 65.4±16.0 | 4.3±3.2 | 62.9±13.2 | 14.1±2.5 | 54.3±27.1 |
| | | MinQ | 12.8±9.3 | 51.8±23.9 | 19.4±11.3 | 60.4±19.4 | 18.0±4.2 | 88.2±11.3 | 8.0±6.7 | 71.1±15.2 | 9.3±8.8 | 71.0±29.1 |
| | m-r 0.10 | U-rand | 31.5±10.6 | 40.3±5.9 | 40.9±2.6 | 46.4±1.8 | 36.9±6.6 | 46.9±1.1 | 38.5±5.7 | 46.8±0.9 | 39.9±2.3 | 61.2±1.1 |
| | | MAD | 19.2±8.2 | 29.4±6.9 | 29.0±2.6 | 46.5±0.9 | 27.1±3.4 | 36.2±0.9 | 22.3±3.8 | 34.5±5.5 | 22.5±1.5 | 62.3±1.0 |
| | | MinQ | 5.1±5.2 | 36.7±8.8 | 39.2±0.8 | 46.2±1.1 | 36.7±6.8 | 44.8±1.1 | 37.0±4.8 | 38.6±1.1 | 34.0±1.4 | 63.2±2.3 |
| Hopper | e 0.05 | U-rand | 46.1±20.7 | 66.9±26.3 | 59.6±29.4 | 95.7±23.8 | 42.6±28.4 | 84.0±27.4 | 53.2±20.8 | 84.4±25.3 | 85.3±37.0 | 81.9±25.2 |
| | | MAD | 31.1±14.4 | 53.2±24.2 | 22.6±13.9 | 73.9±27.9 | 27.2±10.9 | 60.3±27.2 | 36.8±9.0 | 37.1±12.3 | 36.6±22.2 | 59.0±13.8 |
| | | MinQ | 47.4±18.9 | 62.5±27.9 | 32.7±13.5 | 58.7±17.9 | 45.3±27.5 | 95.7±27.6 | 66.7±33.6 | 59.2±23.9 | 79.8±32.7 | 59.4±22.1 |
| | m-r 0.10 | U-rand | 18.5±8.2 | 68.9±19.2 | 66.3±20.1 | 95.9±8.8 | 20.6±9.1 | 65.4±22.0 | 33.9±10.7 | 94.9±17.7 | 80.7±28.0 | 103.5±1.5 |
| | | MAD | 5.1±5.0 | 37.5±26.1 | 32.1±15.9 | 88.9±13.7 | 6.1±5.5 | 64.3±21.8 | 9.9±8.1 | 38.3±15.8 | 51.6±30.7 | 97.5±2.5 |
| | | MinQ | 5.3±5.4 | 18.3±18.4 | 84.6±14.1 | 87.5±6.6 | 11.8±7.6 | 80.5±18.1 | 51.2±25.1 | 62.5±27.3 | 98.3±6.2 | 103.2±2.4 |
| Walker2d | e 0.10 | U-rand | 102.1±1.8 | 110.4±0.8 | 106.1±9.9 | 106.0±7.4 | 106.1±2.9 | 110.0±0.5 | 107.2±1.0 | 109.4±0.5 | 95.1±15.7 | 97.2±9.5 |
| | | MAD | 50.5±43.7 | 70.5±13.3 | 64.1±27.0 | 97.6±16.1 | 19.9±22.7 | 69.7±17.5 | 36.6±35.5 | 88.2±24.8 | 61.9±29.2 | 83.8±19.9 |
| | | MinQ | 99.9±22.2 | 105.6±1.1 | 99.9±11.8 | 102.4±6.9 | 91.9±22.4 | 105.5±1.3 | 101.1±2.0 | 102.4±1.3 | 91.8±28.0 | 89.3±13.3 |
| | m-r 0.15 | U-rand | 17.3±12.2 | 54.9±25.7 | 69.2±20.9 | 78.1±9.2 | 51.2±28.3 | 83.6±14.8 | 64.2±27.8 | 91.1±12.1 | 89.9±1.1 | 88.7±2.1 |
| | | MAD | 6.6±3.3 | 43.4±29.8 | 19.7±14.7 | 78.4±8.8 | 8.8±4.4 | 70.8±19.1 | 7.2±2.3 | 66.1±24.2 | 81.9±11.5 | 90.5±3.5 |
| | | MinQ | 7.3±4.2 | 30.3±26.1 | 66.5±11.8 | 78.5±4.2 | 21.7±15.9 | 76.4±14.9 | 47.2±23.2 | 68.0±19.5 | 82.3±1.4 | 89.6±1.7 |

We consider three additional types of noise attacks that are commonly used on state observations, where DMBP-strengthened algorithms also outperform the corresponding baselines (cf. Table 2):

i) Uniform random noise distributed inside the $\ell_\infty$ ball with the norm of $\kappa$: $\tilde{s}_t = s_t + \kappa \cdot \mathcal{U}(-I, I)$.

ii) Maximum action-difference (adversarial) attack: The noises are selected inside the $\ell_\infty$ ball with the norm of $\kappa$, such that $\tilde{s}_t = s_t + \arg\max_{\tilde{s} \in \mathbb{B}_d(s, \kappa)} D(\pi_\phi(\cdot|s) \| \pi_\phi(\cdot|\tilde{s}))$. Among 20 samples of $\tilde{s}_t$ in the ball, we choose the one with the largest $\|\pi_\phi(\cdot|s) - \pi_\phi(\cdot|\tilde{s})\|^2$.

iii) Minimum Q-value (adversarial) attack: The noises are selected inside the $\ell_\infty$ ball with the norm of $\kappa$ such that, $\tilde{s}_t = s_t + \arg\min_{\tilde{s} \in \mathbb{B}_d(s, \kappa)} Q(\tilde{s}_t, \pi_\phi(\cdot|\tilde{s}))$. Again, we sample 20 times and choose the one with the minimum Q to be the perturbed state $\tilde{s}_t$.

The latter two adversarial attacks have been considered in the literature (Pinto et al., 2017; Zhang et al., 2020; Yang et al., 2022). For fair comparison, when we use DMBP against adversarial attacks, we first sample 20 noised states $\tilde{s}_t$ and denoise them using DMBP, and then choose the denoised states $\hat{s}_t$ with the maximum action difference or the minimum Q-value as the perturbed state.

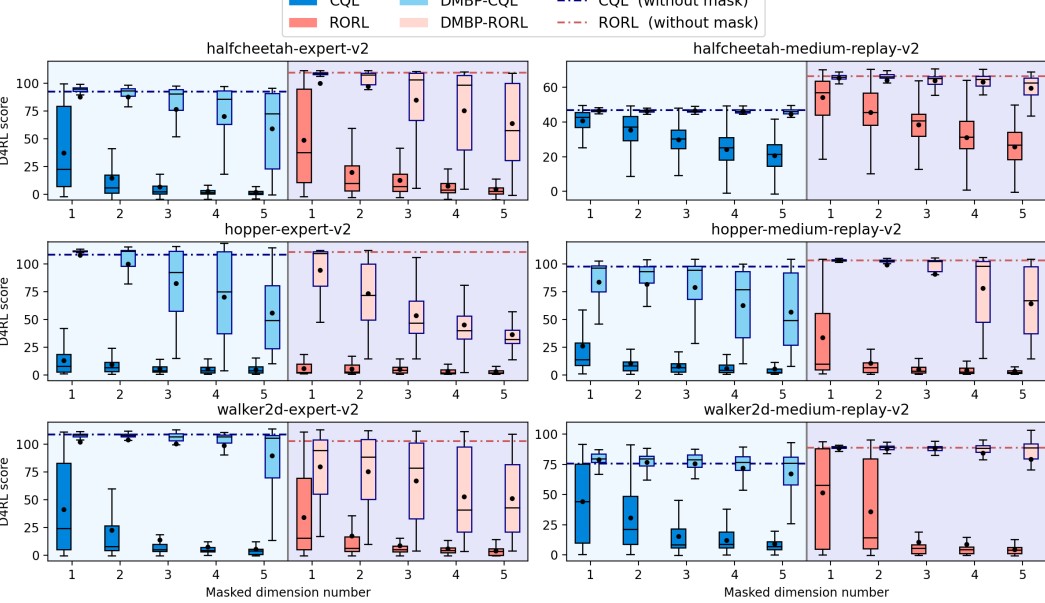

Figure 3: The performance of CQL, DMBP-CQL, RORL, and DMBP-RORL with incomplete state observations that have 1-5 unobserved dimensions. The dash-dot lines represent the performance of the corresponding baseline algorithms in the original environment with fully observable states. (The total state observation dimension is 11 for hopper, and 17 for both halfcheetah and walker2d.)

## 5.2 ROBUSTNESS AGAINST INCOMPLETE STATE OBSERVATIONS WITH UNOBSERVED DIMENSION

We utilize DMBP to recover the missing state information for decision-making. In D4RL benchmark problems, we mask some dimensional state information that cannot be observed by the tested policy (*i.e.*, the masked dimensions of the state are set as 0 for $t \in \{2, 3, \ldots, T\}$). The baseline algorithms make decisions based on the observed (incomplete) states, and DMBP-improved counterparts take actions according to the recovered states. For each dimension of the state, we make the dimension unobserved and conduct 10 tests. When multiple state dimensions cannot be observed, we randomly select 30 groups of dimensions and conduct 10 tests on each group. The experiment results of *CQL*, *RORL* with offline "expert" and "medium-replay" datasets are shown in Figure 3. DMBP significantly enhances the performance of all baseline algorithms by accurately predicting the missing state information. On "medium-replay" datasets, the DMBP strengthened algorithms incur little performance degradation in masked environments, compared with that achieved in the original environments with complete and accurate observations.

## 5.3 ABLATION STUDY

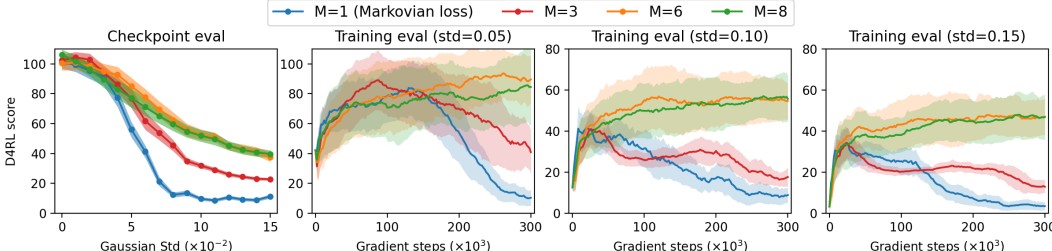

Figure 4: The checkpoint and training process evaluations of DMBP-Diffusion QL under Gaussian random noises, where DMBP is trained on hopper-expert-v2 with different sample trajectory lengths ($M$). The curves are averaged over 5 random seeds, and the checkpoints are selected randomly after gradient steps of $2 \times 10^5$.

In Figure 4, we conduct ablation studies on dataset "hopper-expert-v2", where algorithms are more prone to error accumulation than in other datasets/environments, to demonstrate the efficacy of the proposed non-Markovian loss function and evaluate the impact of the non-Markovian sampling length ($M$ in Eq. 7). We utilize pre-trained Diffusion QL for decision-making to evaluate the performance of DMBP under the framework in Figure 1. Other hyperparameters and DMBP training follow the basic settings in Appendix B.2 and D.1, respectively.

When $M = 1$, the DMBP training objective 7 reduces to the classical training objective of diffusion models in Eq. 1 (*i.e.* $L_M = 0$ in Eq. 7). From the second to the fourth subplots of Figure 4, we observe that the direct adoption of classical conditional diffusion models suffers from severe error accumulation as training proceeds. The proposed non-Markovian training objective significantly enhances the robustness of the baseline RL algorithm against state observation perturbations, especially when the noise scale is large. When $M$ is no less than 6, the performance of DMBP remains almost the same. To expedite the computation, we set $M = 6$ for the "hopper" environment. More ablations studies on neural network structure and condition trajectory lengths ($N$) can be found in Appendix C.

## 6 CONCLUSION

In this work, we propose the first framework of state-denoising for offline RL against observation perturbations in state-based tasks. Leveraging conditional diffusion models, we develop Diffusion Model-Based Predictor (DMBP) to recover the actual state for decision-making. To reduce the error accumulation during test, we propose a new non-Markovian loss function that minimizes the sum entropy of denoised states along the trajectory. Experiments on D4RL benchmarks demonstrate that the proposed DMBP can significantly enhance the robustness of existing offline RL algorithms against different scales of random noises and even adversarial attacks. The proposed framework is shown to be able to effectively deal with the cases of incomplete state observations (with multiple unobserved dimensions) for state-based RL tasks.

ACKNOWLEDGMENTS

This work was supported in part by the General Research Fund (GRF) project 14200720 of the Hong Kong University Grants Committee and the National Natural Science Foundation of China (NSFC) Project 62073273. The authors would like to thank the anonymous reviewers for valuable discussion.

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

## A    DERIVATION OF THE TRAINING OBJECTIVE IN EQ. 7

Following the standard setting of diffusion models with Gaussian distributed noised states (Sohl-Dickstein et al., 2015; Ho et al., 2020), below we derive the non-Markovian training objective via conditional diffusion models.

Suppose that the initial state is perfectly known for the agent, *i.e.*, $s_1$ is known and does not need to be estimated. $\tau_{t-1}^{\hat{s}} := \{\hat{s}_j \mid 1 \le j \le t-1\}$ represents the denoised state trajectory. Our learning objective is to minimize the cross-entropy of the denoised states along the RL trajectory:

$$\mathcal{L}_{\text{entropy}} = \sum_{t=2}^{T} \mathbb{E}_{q(s_t)} \left[ -\log p_\theta(\tilde{s}_t^0 \mid a_{t-1}, \tau_{t-1}^{\hat{s}}) \right].$$

We denote the state in the forward diffusion process as $s_t^i$ and the state in the reverse diffusion process as $\tilde{s}_t^i$. We refer $p_\theta(\tilde{s}_t^0 \mid a_{t-1}, \tau_{t-1}^{\hat{s}})$ to as $p_\theta(\tilde{s}_t^0)$ below. Following Ho et al. (2020), we adopt the variational lower bound (VLB) to optimize the negative log-likelihood:

$$
\begin{aligned}
\mathcal{L}_{\text{VLB}} &= \sum_{t=2}^{T} \mathbb{E}_{q(s_t)} \left[ -\log p_\theta(\tilde{s}_t^0) + D_{KL}(q(s_t^{1:K} \mid s_t^0) \| p_\theta(\tilde{s}_t^{1:K} \mid \tilde{s}_t^0, a_{t-1}, \tau_{t-1}^{\hat{s}}) \right] \\
&= \sum_{t=2}^{T} \left\{ -\mathbb{E}_q \left[ \log p_\theta(\tilde{s}_t^0) \right] + \mathbb{E}_{s_t^{1:K} \sim q(s_t^{1:K} \mid s_t^0)} \left[ \log \frac{q(s_t^{1:K} \mid s_t^0)}{p_\theta(\tilde{s}_t^{1:K} \mid \tilde{s}_t^0, a_{t-1}, \tau_{t-1}^{\hat{s}})} \right] \right\} \\
&= \sum_{t=2}^{T} \left\{ -\mathbb{E}_q \left[ \log p_\theta(\tilde{s}_t^0) \right] + \mathbb{E}_{s_t^{1:K} \sim q(s_t^{1:K} \mid s_t^0)} \left[ \log \frac{q(s_t^{1:K} \mid s_t^0)}{p_\theta(\tilde{s}_t^{0:K} \mid \tilde{s}_t^0, a_{t-1}, \tau_{t-1}^{\hat{s}}) p_\theta(\tilde{s}_t^0)} \right] \right\} \\
&= \sum_{t=2}^{T} \mathbb{E}_{s_t^{1:K} \sim q(s_t^{1:K} \mid s_t^0)} \left[ \log \frac{q(s_t^{1:K} \mid s_t^0)}{p_\theta(\tilde{s}_t^{0:K} \mid \tilde{s}_t^0, a_{t-1}, \tau_{t-1}^{\hat{s}})} \right] \\
&= \sum_{t=2}^{T} \mathbb{E}_{s_t^{1:K} \sim q(s_t^{1:K} \mid s_t^0)} \left[ \log \frac{\prod_{i=1}^{K} q(s_t^i \mid s_t^{i-1})}{p_\theta(\tilde{s}_t^K) \prod_{i=1}^{K} p_\theta(\tilde{s}_t^{i-1} \mid \tilde{s}_t^i, a_{t-1}, \tau_{t-1}^{\hat{s}})} \right] \\
&= \sum_{t=2}^{T} \mathbb{E}_{s_t^{1:K} \sim q(s_t^{1:K} \mid s_t^0)} \left[ -\log p_\theta(\tilde{s}_t^0 \mid \tilde{s}_t^1, a_{t-1}, \tau_{t-1}^{\hat{s}}) + \sum_{i=2}^{K} \log \frac{q(s_t^{i-1} \mid s_t^i, s_t^0)}{p_\theta(\tilde{s}_t^{i-1} \mid \tilde{s}_t^i, a_{t-1}, \tau_{t-1}^{\hat{s}})} \right. \\
&\qquad\qquad\qquad\qquad \left. + \log \frac{q(s_t^K \mid s_t^0)}{p_\theta(\tilde{s}_t^K)} \right] \\
&= \sum_{t=2}^{T} \mathbb{E}_{s_t^{1:K} \sim q(s_t^{1:K} \mid s_t^0)} \left[ -\log p_\theta(\tilde{s}_t^0 \mid \tilde{s}_t^1, a_{t-1}, \tau_{t-1}^{\hat{s}}) \right. \\
&\qquad\qquad\qquad\qquad + \sum_{i=2}^{K} D_{KL} \left( q(s_t^{i-1} \mid s_t^i, s_t^0) \, \| \, p_\theta(\tilde{s}_t^{i-1} \mid \tilde{s}_t^i, a_{t-1}, \tau_{t-1}^{\hat{s}}) \right) \\
&\qquad\qquad\qquad\qquad \left. + D_{KL} \left( q(s_t^K \mid s_t^0) \, \| \, p_\theta(\tilde{s}_t^K) \right) \right],
\end{aligned}
$$

(8)

where $D_{KL}(\cdot \| \cdot)$ represents the Kullback-Leibler divergence and it is guaranteed that $\mathcal{L}_{\text{VLB}} \ge \mathcal{L}_{\text{entropy}}$ (Ho et al., 2020).

As in Ho et al. (2020), for the first term in the last line of Eq. 8, we let

$$p_\theta(\tilde{s}_t^0 \mid \tilde{s}_t^1, a_{t-1}, \tau_{t-1}^{\hat{s}}) = \mathcal{N} \left( \frac{\tilde{s}_t^1}{\sqrt{\alpha_1}} - \frac{\beta_1}{\sqrt{\alpha_1(1 - \bar{\alpha}_1)}} \epsilon_\theta(\tilde{s}_t^1, a_{t-1}, \tau_{t-1}^{\hat{s}}, 1), \beta_1 \Sigma \right). \quad (9)$$

To directly produce the noised state under Gaussian distributed perturbations, we obtain from the forward diffusion process that

$$s_t^i = \sqrt{\bar{\alpha}_i} s_t + \sqrt{1 - \bar{\alpha}_i} \epsilon_t^i, \quad \epsilon_t^i \sim \mathcal{N}(\mathbf{0}, \mathbf{I}), \; i \in \{1, 2, \dots, K\}. \quad (10)$$

For the expression in Eq. 9, we have

$$
\begin{aligned}
&- \log p_\theta(\tilde{s}_t^0 \mid \tilde{s}_t^1, a_{t-1}, \tau_{t-1}^{\hat{s}}) \\
&= \log \left[ (2\pi)^{\frac{d}{2}} \mid \beta_1 \Sigma \mid^{\frac{1}{2}} \right] + \frac{1}{2} \| \tilde{s}_t^0 - \frac{\tilde{s}_t^1}{\sqrt{\alpha_1}} + \frac{\beta_1}{\sqrt{\alpha_1(1-\bar{\alpha}_1)}} \epsilon_\theta(\tilde{s}_t^1, a_{t-1}, \tau_{t-1}^{\hat{s}}, 1) \|_{(\beta_1 \Sigma)^{-1}}^2 \quad (11) \\
&= \frac{1}{2} \log \left[ (2\pi)^d \mid \beta_1 \Sigma \mid \right] + \frac{1}{2\alpha_1} \| \epsilon_\theta(\tilde{s}_t^1, a_{t-1}, \tau_{t-1}^{\hat{s}}, 1) - \epsilon_t^{i=1} \|_{(\Sigma)^{-1}}^2,
\end{aligned}
$$

where the first equality follows from Eq. 9, and the second equality holds due to Eq. 10.

For the second term in the last line of Eq. 8, we let (Ho et al., 2020)

$$
q(s_t^{i-1} \mid s_t^i, s_t^0) = \mathcal{N} \left( \frac{\sqrt{\alpha_i}(1-\bar{\alpha}_{i-1})}{1-\bar{\alpha}_i} s_t^i + \frac{\sqrt{\bar{\alpha}_{i-1}}\beta_i}{1-\bar{\alpha}_i} s_t^0, \, \frac{1-\bar{\alpha}_{i-1}}{1-\bar{\alpha}_i}\beta_i \Sigma \right), \quad (12)
$$

and

$$
p_\theta(\tilde{s}_t^{i-1} \mid \tilde{s}_t^i, a_{t-1}, \tau_{t-1}^{\hat{s}}) = \mathcal{N} \left( \frac{\tilde{s}_t^i}{\sqrt{\alpha_i}} - \frac{\beta_i}{\sqrt{\alpha_i(1-\bar{\alpha}_i)}} \epsilon_\theta(\tilde{s}_t^i, a_{t-1}, \tau_{t-1}^{\hat{s}}, i), \, \frac{1-\bar{\alpha}_{i-1}}{1-\bar{\alpha}_i}\beta_i \Sigma \right). \quad (13)
$$

Utilizing the Kullback-Leibler divergence for Gaussian distributions in Eq. 12 and Eq. 13, we obtain

$$
\begin{aligned}
&D_{KL} \left( q(s_t^{i-1} \mid s_t^i, s_t^0) \, \| \, p_\theta(\tilde{s}_t^{i-1} \mid \tilde{s}_t^i, a_{t-1}, \tau_{t-1}^{\hat{s}}) \right) \\
&= \frac{1}{2} \| \frac{\sqrt{\alpha_i}(1-\bar{\alpha}_{i-1})}{1-\bar{\alpha}_i} s_t^i + \frac{\sqrt{\bar{\alpha}_{i-1}}\beta_i}{1-\bar{\alpha}_i} s_t^0 - \frac{\tilde{s}_t^i}{\sqrt{\alpha_i}} + \frac{\beta_i}{\sqrt{\alpha_i(1-\bar{\alpha}_i)}} \epsilon_\theta(\tilde{s}_t^i, a_{t-1}, \tau_{t-1}^{\hat{s}}, i) \|_{\left(\frac{1-\bar{\alpha}_{i-1}}{1-\bar{\alpha}_i}\beta_i \Sigma\right)^{-1}}^2 \\
&= \frac{\beta_i}{2\alpha_i(1-\bar{\alpha}_{i-1})} \| \epsilon_\theta(\tilde{s}_t^i, a_{t-1}, \tau_{t-1}^{\hat{s}}, i) - \epsilon_t^i \|_{\Sigma^{-1}}^2,
\end{aligned}
$$
$$(14)$$

where the last equality follows from Eq. 10.

As in (Ho et al., 2020), for the third term in the last line of Eq. 8, we let

$$
q(s_t^K \mid s_t^0) = \mathcal{N} \left( \sqrt{\bar{\alpha}_K} s_t^0, \, (1-\bar{\alpha}_K)\Sigma \right). \quad (15)
$$

In a similar spirit to the pure Gaussian distributed noise assumption made in Ho et al. (2020), we assume that the observed noised state follows a Gaussian distribution, *i.e.*,

$$
p_\theta(\tilde{s}_t^K) = \mathcal{N} \left( m s_t^0, \, n\Sigma \right). \quad (16)
$$

According to Eq. 15 and Eq. 16, we obtain the KL divergence of two Gaussian distributions

$$
D_{KL} \left( q(s_t^K \mid s_t^0) \, \| \, p_\theta(\tilde{s}_t^K) \right) = \frac{1}{2} \left[ \| (\sqrt{\bar{\alpha}_K} - m)s_t^0 \|_{(n\Sigma)^{-1}}^2 + d\frac{1-\bar{\alpha}_K}{n} - d - d\log\frac{1-\bar{\alpha}_K}{n} \right], \quad (17)
$$

which is a constant and does not depend on the neural network parameters $\epsilon_\theta$.

Substituting Eqs. 11 14 17 to the last line in Eq. 8, we can express $\mathcal{L}_{\text{VLB}}$ as

$$
\begin{aligned}
\mathcal{L}_{\text{VLB}} &= \sum_{t=2}^T \mathbb{E}_{s_1 \sim d_0, \epsilon_t^i \sim \mathcal{N}(0,I)} \left[ C_t + \sum_{i=1}^K \gamma^i \| \epsilon_\theta(\tilde{s}_t^i, a_{t-1}, \tau_{t-1}^{\hat{s}}, i) - \epsilon_t^i \|_{\Sigma^{-1}}^2 \right] \\
&= \mathbb{E}_{s_1 \sim d_0, \epsilon_t^i \sim \mathcal{N}(0,I)} \left[ \sum_{t=2}^T C_t + \sum_{t=2}^T \sum_{i=1}^K \gamma^i \| \epsilon_\theta(\tilde{s}_t^i, a_{t-1}, \tau_{t-1}^{\hat{s}}, i) - \epsilon_t^i \|_{\Sigma^{-1}}^2 \right],
\end{aligned}
$$
$$(18)$$

where $C_t$ is a constant for $t \in \{1, 2, \ldots, T-1\}$:

$$
C_t = \frac{1}{2} \left[ \| (\sqrt{\bar{\alpha}_K} - m)s_t^0 \|_{(n\Sigma)^{-1}}^2 + d\frac{1-\bar{\alpha}_K}{n} - d - d\log\frac{1-\bar{\alpha}_K}{n} + \log \left[ (2\pi)^d \mid \beta_1 \Sigma \mid \right] \right],
$$

and the coefficients for the norm terms are given by

$$\gamma^i = \begin{cases} \dfrac{1}{2\alpha_1} & \text{if } i = 1, \\ \dfrac{\beta_i}{2\alpha_i(1 - \bar{\alpha}_{i-1})} & \text{otherwise } (i \in \{2, 3, \ldots, K\}). \end{cases}$$

Following the setting in Ho et al. (2020), we ignore the constant term and the coefficients. The training objective in Eq. 18 can be simplified as

$$\mathcal{L}_{\text{simple}}(\theta) = \mathbb{E}_{\boldsymbol{s}_1 \sim d_0, \boldsymbol{\epsilon}_t^i \sim \mathcal{N}(\mathbf{0}, \boldsymbol{I})} \left[ \sum_{t=2}^{T} \sum_{i=1}^{K} \|\boldsymbol{\epsilon}_\theta(\tilde{\boldsymbol{s}}_t^i, \boldsymbol{a}_{t-1}, \boldsymbol{\tau}_{t-1}^{\hat{\boldsymbol{s}}}, i) - \boldsymbol{\epsilon}_t^i\|^2 \right]$$

$$= \mathbb{E}_{\boldsymbol{s}_1 \sim d_0, \boldsymbol{\epsilon}_t^i \sim \mathcal{N}(\mathbf{0}, \boldsymbol{I})} \left[ \sum_{i=1}^{K} \sum_{t=2}^{T} \|\boldsymbol{\epsilon}_\theta(\tilde{\boldsymbol{s}}_t^i, \boldsymbol{a}_{t-1}, \boldsymbol{\tau}_{t-1}^{\hat{\boldsymbol{s}}}, i) - \boldsymbol{\epsilon}_t^i\|^2 \right] \qquad (19)$$

$$= \mathbb{E}_{\boldsymbol{s}_1 \sim d_0, \boldsymbol{\epsilon}_t^i \sim \mathcal{N}(\mathbf{0}, \boldsymbol{I}), i \sim \mathcal{U}_K} \left[ \sum_{t=2}^{T} \|\boldsymbol{\epsilon}_\theta(\tilde{\boldsymbol{s}}_t^i, \boldsymbol{a}_{t-1}, \boldsymbol{\tau}_{t-1}^{\hat{\boldsymbol{s}}}, i) - \boldsymbol{\epsilon}_t^i\|^2 \right],$$

where $\mathcal{U}_K$ is the uniform distribution over discrete set $\{1, 2, \ldots, K\}$, and for $\boldsymbol{\tau}_{t-1}^{\hat{\boldsymbol{s}}} := \{\hat{\boldsymbol{s}}_j \mid j \leq t-1\}$, we have

$$\hat{\boldsymbol{s}}_j = \begin{cases} \boldsymbol{s}_1 & \text{if } j = 1, \\ f_k(\tilde{\boldsymbol{s}}_j^k) \displaystyle\prod_{i=1}^{k} p_\theta(\tilde{\boldsymbol{s}}_j^{i-1} \mid \tilde{\boldsymbol{s}}_j^i, \boldsymbol{a}_{j-1}, \boldsymbol{\tau}_{j-1}^{\hat{\boldsymbol{s}}}) & \text{otherwise } (j \in \{2, \ldots, t-1\}). \end{cases}$$

From 19, it is clear that the simplified target is to make $\boldsymbol{\epsilon}_\theta(\tilde{\boldsymbol{s}}_t^i, \boldsymbol{a}_{t-1}, \boldsymbol{\tau}_{t-1}^{\hat{\boldsymbol{s}}}, i)$ as close as possible to $\boldsymbol{\epsilon}_t^i$ for any $t$. From the forward noising process in Eq. 10, we have the following approximation

$$\hat{\boldsymbol{s}}_t = \tilde{\boldsymbol{s}}_t^{0(i)} \approx \frac{\tilde{\boldsymbol{s}}_t^i}{\sqrt{\bar{\alpha}_i}} - \frac{\sqrt{\bar{\alpha}_i}}{\sqrt{1 - \bar{\alpha}_i}} \boldsymbol{\epsilon}_\theta(\tilde{\boldsymbol{s}}_t^i, \boldsymbol{a}_{t-1}, \boldsymbol{\tau}_{t-1}^{\hat{\boldsymbol{s}}}, i), \quad t \in \{2, \ldots, T-1\}. \qquad (20)$$

Therefore, Eq. 19 can be expressed as:

$$\mathcal{L}_{\text{simple}}(\theta) = \mathbb{E}_{\boldsymbol{s}_1 \sim d_0, \boldsymbol{\epsilon}_t^i \sim \mathcal{N}(\mathbf{0}, \boldsymbol{I}), i \sim \mathcal{U}_K} \Big[ \underbrace{\|\boldsymbol{\epsilon}_\theta(\tilde{\boldsymbol{s}}_2^i, \boldsymbol{a}_1, \boldsymbol{s}_1, i) - \boldsymbol{\epsilon}_2^i\|^2}_{\text{term1}} +$$

$$\underbrace{\sum_{t=3}^{T} \|\boldsymbol{\epsilon}_\theta(\tilde{\boldsymbol{s}}_t^i, \boldsymbol{a}_{t-1}, \boldsymbol{\tau}_{t-1}^{\hat{\boldsymbol{s}}}, i) - \boldsymbol{\epsilon}_t^i\|^2}_{\text{term2}} \Big], \qquad (21)$$

where $\boldsymbol{\tau}_{t-1}^{\hat{\boldsymbol{s}}} := \{\hat{\boldsymbol{s}}_j \mid j \leq t-1\}$, and $\hat{\boldsymbol{s}}_t$ along the RL trajectory can be calculated through Eq. 20 iteratively from $t = 2$ to $t = T$. However, this is time-consuming as the sampling process needs to be looped for $(T - 2)$ times. In addition, the historical state trajectories multiple RL timesteps ago have little impact on the current denoising results. To expedite the computation, we extract a partial trajectory with a window size of $N + M$ ($N$ is the condition trajectory length and $M$ is the sample trajectory length) in Eq. 21, and set the training objective as in Eq. 7:

$$\mathcal{L}(\theta) = \mathbb{E}_{i \sim \mathcal{U}_K, \boldsymbol{\epsilon}_t \sim \mathcal{N}(\mathbf{0}, \boldsymbol{I}), (\boldsymbol{s}_{t-N}, \ldots, \boldsymbol{s}_{t+M-1}) \in \mathcal{D}_\nu} \Bigg[ \underbrace{\|\boldsymbol{\epsilon}_\theta(\tilde{\boldsymbol{s}}_t^i, \boldsymbol{a}_{t-1}, \boldsymbol{\tau}_{t-1}^{\boldsymbol{s}}, i) - \boldsymbol{\epsilon}_t^i\|^2}_{L_t} +$$

$$\underbrace{\sum_{m=t+1}^{t+M-1} \|\boldsymbol{\epsilon}_\theta(\tilde{\boldsymbol{s}}_m^i, \boldsymbol{a}_{m-1}, \boldsymbol{\tau}_{m-1}^{\check{\boldsymbol{s}}}, i) - \boldsymbol{\epsilon}_m^i\|^2}_{L_m} \Bigg],$$

where the state trajectory condition for the predictor $\boldsymbol{\epsilon}_\theta$ in $L_t$ is the original $\boldsymbol{\tau}_{t-1}^{\boldsymbol{s}} = \{\boldsymbol{s}_{t-N}, \ldots, \boldsymbol{s}_{t-1}\}$ from the offline dataset $\mathcal{D}_\nu$, and the state trajectory condition in $L_m$ can be expressed as $\boldsymbol{\tau}_{m-1}^{\check{\boldsymbol{s}}} = \{\check{\boldsymbol{s}}_j \mid j \in \{m - N, \ldots, m - 1\}\}$, with:

$$\check{\boldsymbol{s}}_j = \begin{cases} \boldsymbol{s}_j & \text{if } j < t, \\ \dfrac{1}{\sqrt{\bar{\alpha}_i}} \big[ \tilde{\boldsymbol{s}}_j^i - \sqrt{1 - \bar{\alpha}_i} \boldsymbol{\epsilon}_\theta(\tilde{\boldsymbol{s}}_j^i, \boldsymbol{a}_{j-1}, \boldsymbol{\tau}_{j-1}^{\check{\boldsymbol{s}}}, i) \big] & \text{otherwise } (j \in \{t, \ldots, t + M - 2\}). \end{cases}$$

# B  IMPLEMENTATION DETAILS

## B.1  DMBP NETWORK STRUCTURE

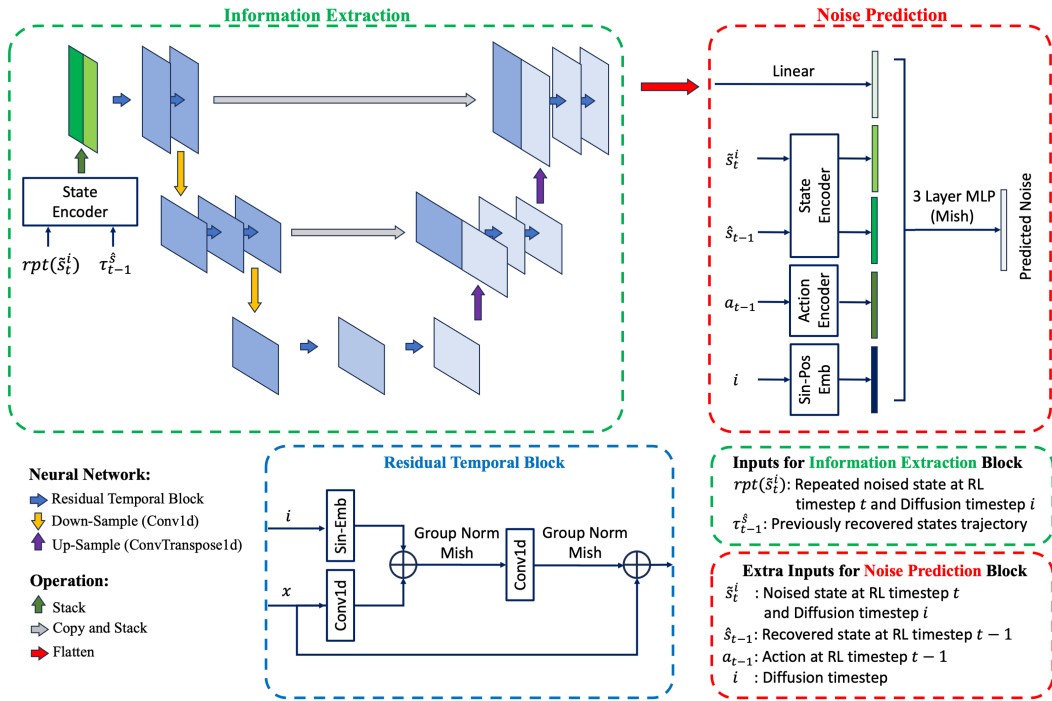

Figure 5: Neural network structure of DMBP.

We implement DMBP based on classical diffusion models (Ho et al., 2020). As discussed in Section 4.1, the naive adoption of MLP neural network structure, which stacks all inputs in one dimension, leads to poor estimation of noise, partially due to the fact that $\hat{s}_{t-1}$ is more closely related to $\tilde{s}_t^i$ than $\hat{s}_j$ with $j < t - 1$, and that this information cannot be well captured by the MLP structure. Therefore, we first extract the information from the trajectory with U-net (Ronneberger et al., 2015; Janner et al., 2022) (see "Information Extraction" block in Figure 5). Then, the output of U-net $U_\xi(\tilde{s}_t^i, \tau_{t-1}^{\hat{s}})$ is flattened on one dimension and fed into "Noise Prediction" block together with the noised current state $\tilde{s}_t^i$, last-step agent generated action $a_{t-1}$, last-step denoised state $\hat{s}_{t-1}$, and the diffusion timestep $i$.

It should be mentioned that directly inputting state and action to neural networks may lead to poor estimation performance, partially due to the much higher dimensionality of state than that of action. We utilize MLP-based neural networks as encoders to match the dimensions of action and states. The final output of our Unet-MLP neural network is recorded as $\epsilon_\theta(\tilde{s}_t^i, a_{t-1}, \tau_{t-1}^{\hat{s}}, i)$. Detailed hyperparameter settings can be found in Appendix B.2, and ablation studies on network structure can be found in Appendix C.

## B.2  DMBP HYPERPARAMETERS AND TRAINING

As discussed in Section 4.1, the proposed framework essentially deals with information with small to medium scale noises instead of generating data from pure noise. We redesign the variance schedule to restrict the noise scale in the diffusion process and limit the total number of diffusion timesteps $K$ for predictor training. The generic neural network and the variance schedule hyperparameters can be found in Table 3.

Since action and state dimensions vary greatly in different environments, we choose different numbers of neurons in the proposed MLP-based encoders in various environments. The corresponding hyperparameter is recorded as the embedded dimension $\zeta$, and the action/state encoder can be

expressed as FC($2\zeta$, $\frac{1}{2}\zeta$) with Mish activations. Besides, for different benchmark environments and datasets, we choose different condition trajectory lengths ($N$) and sample trajectory lengths ($M$) to achieve the best performance (cf. Table 4).

Table 3: Generic hyperparameters of DMBP

| Hyper-parameters | Value |
|---|---|
| Noise prediction network | FC(256,256,256) with Mish activations |
| Dropout for predictor network | 0.1 |
| Learning rate | 3e-4 |
| Batch size | 64 |
| Variance schedule | $a = 3.065$, $b = 24.552$, $c = -3.170$ |
| Total diffusion step ($K$) | 10 for denoising tasks |
| | 100 for infilling tasks |

Table 4: Hyperparameters of DMBP for different benchmark environments and datasets

| Domain | Environment | Dataset | Embedded dimension ($\zeta$) | Condition trajectory length ($N$) | Sample trajectory length ($M$) |
|---|---|---|---|---|---|
| Mujoco | halfcheetah | all | 64 | 4 | 2 |
| | hopper | all | 64 | 4 | 6 |
| | walker2d | all | 64 | 4 | 4 |
| Adroit | pen | "expert" | 128 | 4 | 6 |
| | hammer | "expert" | 128 | 4 | 6 |
| | door | "expert" | 128 | 4 | 6 |
| | relocate | "expert" | 128 | 4 | 4 |
| Franka Kitchen | kitchen | "mixed" | 256 | 4 | 4 |
| | kitchen | "complete" | 256 | 4 | 4 |
| | kitchen | "partial" | 256 | 4 | 4 |

Different from classical conditional diffusion models, our proposed DMBP utilizes a non-Markovian loss function 7 for updating. For each iteration, we sample the partial trajectory $(\boldsymbol{s}_{t-N}, \boldsymbol{a}_{t-N}, \boldsymbol{s}_{t-N+1}, \dots, \boldsymbol{s}_{t+M-1})$ from offline dataset $\mathcal{D}_\nu$. Following the setting in Section 4.2, we present the procedure for DMBP training in Algorithm 2.

---

**Algorithm 2** DMBP algorithm

**Require:** offline dataset $\mathcal{D}_v$, initialized noise predictor $\boldsymbol{\epsilon}_\theta$

1: **for** each iteration **do**
2:     Sample trajectory mini-batch $\mathcal{B} = \{(\boldsymbol{s}_{t-N}, \boldsymbol{a}_{t-N}, \boldsymbol{s}_{t-N+1}, \dots, \boldsymbol{s}_{t+M-1})\} \sim \mathcal{D}_v$
3:     Sample uniformly distributed diffusion timestep $i \sim \{1, 2, \dots, K\}$
4:     Sample random Gaussian noise $\boldsymbol{\epsilon}_t^i \sim \mathcal{N}(\boldsymbol{0}, \boldsymbol{I})$
5:     Produce noised state through $\tilde{\boldsymbol{s}}_t^i = \sqrt{\bar{\alpha}_i}\boldsymbol{s}_t + \sqrt{1-\bar{\alpha}_i}\boldsymbol{\epsilon}_t^i$
6:     Get Trajectory $\boldsymbol{\tau}_{t-1}^{\hat{\boldsymbol{s}}} = \{\boldsymbol{s}_{t-N}, \dots, \boldsymbol{s}_{t-1}\}$
7:     $L_t = \|\boldsymbol{\epsilon}_\theta(\tilde{\boldsymbol{s}}_t^i, \boldsymbol{a}_{t-1}, \boldsymbol{\tau}_{t-1}^{\hat{\boldsymbol{s}}}, i) - \boldsymbol{\epsilon}_t^i\|^2$
8:     Recover the noised state through $\breve{\boldsymbol{s}}_t = \frac{1}{\sqrt{\bar{\alpha}_i}}\left[\tilde{\boldsymbol{s}}_t^i - \sqrt{1-\bar{\alpha}_i}\boldsymbol{\epsilon}_\theta(\tilde{\boldsymbol{s}}_t^i, \boldsymbol{a}_{t-1}, \boldsymbol{\tau}_{t-1}^{\hat{\boldsymbol{s}}}, i)\right]$
9:     $\boldsymbol{\tau}_t^{\hat{\boldsymbol{s}}} = \text{POP}(\text{PUSH}(\boldsymbol{\tau}_{t-1}^{\hat{\boldsymbol{s}}}, \breve{\boldsymbol{s}}_t), \boldsymbol{s}_{t-N})$
10:     **for** $m = t+1, \dots, t+M-1$ **do**
11:         Sample $\boldsymbol{\epsilon}_m^i \sim \mathcal{N}(\boldsymbol{0}, \boldsymbol{I})$
12:         $\tilde{\boldsymbol{s}}_m^i = \sqrt{\bar{\alpha}_i}\boldsymbol{s}_m + \sqrt{1-\bar{\alpha}_i}\boldsymbol{\epsilon}_m^i$
13:         $L_m = \|\boldsymbol{\epsilon}_\theta(\tilde{\boldsymbol{s}}_m^i, \boldsymbol{a}_{m-1}, \boldsymbol{\tau}_{m-1}^{\hat{\boldsymbol{s}}}, i) - \boldsymbol{\epsilon}_m^i\|^2$
14:         $\breve{\boldsymbol{s}}_m = \frac{1}{\sqrt{\bar{\alpha}_i}}\left[\tilde{\boldsymbol{s}}_m^i - \sqrt{1-\bar{\alpha}_i}\boldsymbol{\epsilon}_\theta(\tilde{\boldsymbol{s}}_m^i, \boldsymbol{a}_{m-1}, \boldsymbol{\tau}_{m-1}^{\hat{\boldsymbol{s}}}, i)\right]$
15:         $\boldsymbol{\tau}_m^{\hat{\boldsymbol{s}}} = \text{POP}(\text{PUSH}(\boldsymbol{\tau}_{m-1}^{\hat{\boldsymbol{s}}}, \breve{\boldsymbol{s}}_m), \breve{\boldsymbol{s}}_{m-N})$
16:     **end for**
17:     Update noise predictor $\boldsymbol{\epsilon}_\theta$ by minimizing $\sum_{n=t}^{t+M-1} L_n$
18: **end for**
19: **return** $\boldsymbol{\epsilon}_\theta$

---

## C  ADDITIONAL ABLATION STUDIES

As introduced in Section 5.3, our proposed non-Markovian training objective is essential for avoiding error accumulation in model-based estimation. To demonstrate the efficacy of the proposed approach, we visualize partial trajectory of "hopper" during the test in Figure 6.

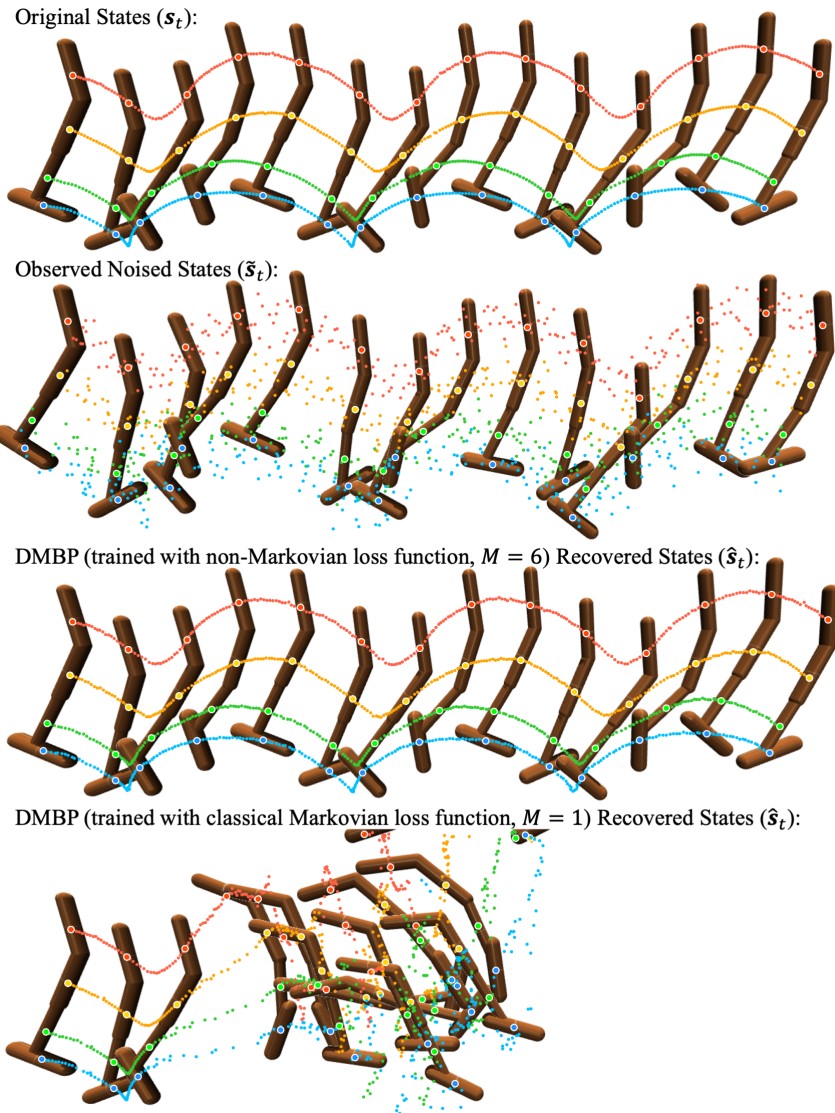

Figure 6: Visualization of the denoising effect of DMBP (trained with different sample trajectory length $M$) alongside Diffusion QL (all trained on the dataset hopper-expert-v2). During the test, the observation is perturbed with Gaussian distributed random noise with std of 0.10, and the action is determined based on the denoising results of DMBP trained with $M = 6$.

When the proposed non-Markovian loss function is adopted to train DMBP (see the third subplot of Figure 6), DMBP can give accurate predictions on the original states in a long RL trajectory. Even though prediction errors exist for some RL timesteps, such errors do not collapse future prediction results. However, when the classical Markovian loss function is used to train DMBP (in the fourth subplot of Figure 6), prediction error accumulates, leading to severe prediction errors and significant deviation of the predicted trajectory (away from the actual one).

We further conduct ablation studies on the condition trajectory length ($N$) in Figure 7. When $N = 1$, the noise prediction $\epsilon_\theta(\tilde{s}_t^i, a_{t-1}, \tau_{t-1}^{\hat{s}}, i)$ decays to $\epsilon_\theta(\tilde{s}_t^i, a_{t-1}, \hat{s}_{t-1}, i)$, *i.e.*, the state condition for DMBP reduces to only the last step denoised state $\hat{s}_{t-1}$ (instead of the trajectory $\tau_{t-1}^{\hat{s}}$). Theoretically, this is enough for predicting the noise information as reinforcement learning problems are always modeled as Markov decision processes (MDP), and most model-based methods follow this assumption (Lin et al., 2017; Janner et al., 2019; Yu et al., 2020; 2021). However, we observe that DMBP performs poorly in the conventional MDP training setting with $N = 1$. On the contrary, larger condition trajectory length $N$ leads to better prediction of the noised states. This is partially due to the fact that when predicting the current state ($\hat{s}_t$), the prediction error of the previous RL timestep denoised state ($\hat{s}_{t-1}$) can be compensated by the information extracted on the state trajectory ($\tau_{t-1}^{\hat{s}}$). We set $N = 4$ for the "hopper" environment.

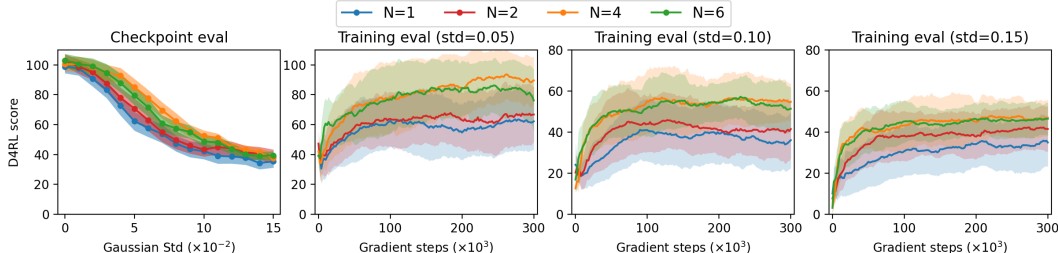

Figure 7: Ablation studies of condition trajectory length ($N$) on dataset "hopper-expert-v2". $M$ is set to be 6 for all DMBP.

Lastly, we conduct the ablation studies on our proposed Unet-MLP structure. As discussed in Section 4.1, directly inputting state trajectories and action into MLP networks leads to poor noise estimation. In Figure 8, we compare the performance of DMBP with MLP network structure and DMBP with the proposed Unet-MLP network structure. For the MLP network, we stack all information in one dimension as input, and the network hyperparameters are set to be the same as the "Noise Prediction" module of our Unet-MLP structure (cf. Figure 5). For the Unet-MLP network, we follow the basic settings in Appendix B. We observe that the MLP network converges faster but achieves much worse denoising performance than Unet-MLP.

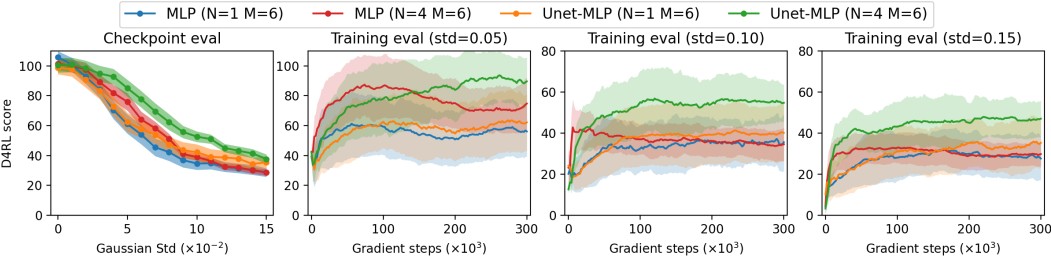

Figure 8: Ablation studies of the proposed Unet-MLP structure on dataset "hopper-expert-v2".

## D  EXPERIMENTAL DETAILS

### D.1  EXPERIMENT SETTINGS

For the baseline RL algorithms (including *BCQ* (Fujimoto et al., 2019), *CQL* (Kumar et al., 2020), *TD3+BC* (Fujimoto & Gu, 2021), *Diffusion QL* (Wang et al., 2022), and *RORL* (Yang et al., 2022)), we train them with the suggested hyperparameters for specific environments and datasets. For the sake of fair comparison, we set checkpoints every 10 epochs (1000 gradient steps for each epoch) during the training process of each algorithm, and randomly select 5 from the last 10 checkpoint policies for robustness evaluation.

As for the training process of DMBP, we follow the hyperparameter settings in Appendix B.2. We train DMBP for 300 epochs and set the checkpoints every 10 epochs. Analogously, we randomly select 5 from the last 10 checkpoint DMBPs and pair them randomly with the previously selected policies for robustness evaluation.

When DMBP is adopted to improve the robustness against noised state observations (denoising tasks introduced in Section 5.1), we utilize different diffusion start timesteps ($k$) (see Eq. 2) for different types and scales of noises (cf. Table 5). When DMBP is used for improving the robustness of policy against incomplete state observations with unobserved dimension(s) (the infilling tasks introduced in Section 5.2), we set the diffusion start timestep to be 100 and the resampling times to be 2 for all benchmark experiments.

Table 5: Choices of the diffusion start timestep under different types and scales of noises

| Gaussian | | Uniform | | Adversarial | |
|---|---|---|---|---|---|
| Noise Scale (std) | Diffusion start timestep ($k$) | Noise Scale (norm) | Diffusion start timestep ($k$) | Noise Scale (norm) | Diffusion start timestep ($k$) |
| (0.00, 0.03] | 1 | (0.00, 0.05] | 1 | (0.00, 0.04] | 1 |
| (0.03, 0.06] | 2 | | | (0.04, 0.08] | 2 |
| (0.06, 0.09] | 3 | (0.05, 0.10] | 2 | (0.08, 0.12] | 3 |
| (0.09, 0.12] | 4 | (0.10, 0.15] | 3 | (0.12, 0.15] | 4 |
| (0.12, 0.15] | 5 | | | | |

## D.2 ADDITIONAL EXPERIMENTAL RESULTS ON MUJOCO

We provide more experimental results in addition to the robustness evaluation of DMBP against noised state observation (in Section 5.1) and against incomplete state observations with unobserved dimensions (in Section 5.2).

Following the basic setting with Gaussian noises defined in Section 5.1, the full robustness evaluation results of five baseline algorithms (*BCQ*, *CQL*, *TD3+BC*, *Diffusion QL*, and *RORL*) on the Mujoco domain with and without DMBP are presented in Table 6, where we include the training datasets "medium-expert", "medium", and "full-replay". In Table 7, we further provide full evaluation results of three other types of noise attacks that have been commonly adopted on state observations (including uniform random noise, maximum action-difference attack, and minimum Q-value attack). It is clear that DMBP greatly enhances the robustness of all baseline algorithms against different types of attacks on state observation for all environments and datasets. Almost all baseline algorithms (without robustness consideration) strengthened by DMBP outperform the SOTA robust offline RL algorithm *RORL*.

To better demonstrate the robustness enhancement of DMBP on the baseline algorithms under different noise scales, we present the robustness evaluation of *CQL* and *RORL* against four different attacks in Figure 9.

For the robustness evaluation against incomplete state observations with unobserved dimensions (see experiment settings in Section 5.2), we perform the robustness evaluation test under three baseline RL algorithms, *BCQ*, *TD3+BC*, and *Diffusion QL*, on halfcheetah (cf. Figure 10), hopper (cf. Figure 11), and walker2d (cf. Figure 12); the training datasets are chosen to be "medium-expert", "medium" and "full-replay". The evaluation results highlight the robust adaptability of DMBP, *i.e.*, its compatibility with a wide range of offline RL algorithms and diverse datasets.

Table 6: D4RL score of the baseline algorithms and the DMBP strengthened ones trained with expert (e), medium-expert (m-e), medium (m), medium-replay (m-r), and full-replay (f-r) datasets in Mujoco domain under different scales of Gaussian random noise on state observation.

| Env Dataset | | Noise scale | BCQ base | BCQ DMBP | CQL base | CQL DMBP | TD3+BC base | TD3+BC DMBP | Diffusion QL base | Diffusion QL DMBP | RORL base | RORL DMBP |
|---|---|---|---|---|---|---|---|---|---|---|---|---|
| HalfCheetah | e | 0 | 96.9±1.8 | - | 93.0±6.1 | - | 95.8±8.9 | - | 92.9±10.7 | - | 108.5±11.2 | - |
| | | 0.05 | 4.5±2.6 | 60.2±23.9 | 18.1±8.6 | 60.9±22.5 | 7.3±6.6 | 77.1±15.5 | 4.8±3.6 | 75.2±20.7 | 15.4±3.9 | 55.7±29.2 |
| | | 0.10 | 4.5±2.5 | 26.8±16.2 | 7.4±4.0 | 40.5±16.6 | 4.7±3.6 | 47.5±22.2 | 3.3±2.5 | 39.8±21.8 | 3.7±1.9 | 32.8±20.4 |
| | m-e | 0 | 94.5±3.8 | - | 92.0±10.6 | - | 89.0±12.1 | - | 94.4±1.5 | - | 106.5±3.6 | - |
| | | 0.05 | 38.9±5.3 | 64.8±17.9 | 34.5±9.8 | 66.2±19.6 | 38.8±4.2 | 72.6±12.7 | 38.0±7.4 | 80.1±9.9 | 55.2±17.3 | 69.8±15.0 |
| | | 0.10 | 28.9±4.2 | 48.7±9.9 | 26.8±5.3 | 43.6±16.1 | 26.9±5.4 | 52.1±16.7 | 28.5±3.8 | 63.5±10.3 | 21.5±6.5 | 46.2±9.7 |
| | m | 0 | 47.6±0.9 | - | 49.4±0.9 | - | 48.4±0.8 | - | 52.5±0.7 | - | 64.2±3.3 | - |
| | | 0.10 | 29.3±5.9 | 46.2±1.1 | 31.7±6.2 | 47.7±1.1 | 28.9±1.8 | 47.0±0.8 | 29.5±5.0 | 48.1±1.6 | 43.8±9.5 | 59.2±4.2 |
| | | 0.15 | 22.1±3.8 | 43.6±4.1 | 24.6±5.8 | 46.3±2.6 | 21.9±5.9 | 45.4±1.1 | 21.7±5.8 | 45.5±6.3 | 30.5±7.6 | 57.9±6.8 |
| | m-r | 0 | 41.6±4.2 | - | 47.0±1.0 | - | 45.2±0.9 | - | 47.7±0.8 | - | 66.7±1.4 | - |
| | | 0.10 | 20.6±6.9 | 38.5±11.2 | 35.6±1.3 | 45.8±1.0 | 28.5±5.5 | 44.3±1.0 | 30.1±4.1 | 45.6±0.9 | 43.5±2.4 | 61.9±1.2 |
| | | 0.15 | 14.8±10.2 | 35.1±8.7 | 28.8±1.5 | 44.6±1.1 | 24.0±8.9 | 42.5±2.6 | 24.2±7.6 | 44.6±3.0 | 30.3±5.9 | 58.4±1.2 |
| | f-r | 0 | 75.5±1.8 | - | 78.7±2.3 | - | 75.7±1.5 | - | 77.9±1.3 | - | 88.1±3.0 | - |
| | | 0.10 | 27.4±7.4 | 59.8±4.4 | 31.8±3.2 | 63.4±2.7 | 27.9±8.3 | 58.3±8.4 | 30.7±3.3 | 62.8±2.1 | 43.2±6.5 | 67.5±5.8 |
| | | 0.15 | 21.2±6.9 | 48.4±12.4 | 25.0±2.0 | 55.8±2.4 | 23.7±3.8 | 51.9±6.3 | 22.4±4.9 | 55.1±2.5 | 33.9±4.6 | 62.3±7.9 |
| Hopper | e | 0 | 88.4±22.4 | - | 109.1±13.7 | - | 108.9±10.5 | - | 104.9±15.1 | - | 110.4±3.1 | - |
| | | 0.05 | 34.3±13.4 | 61.0±25.2 | 41.2±21.8 | 85.7±26.2 | 32.2±18.4 | 79.1±28.2 | 38.2±12.4 | 84.8±27.4 | 56.9±34.9 | 64.3±19.8 |
| | | 0.10 | 24.3±10.9 | 37.1±18.5 | 24.3±11.8 | 48.8±20.4 | 22.7±11.6 | 32.6±18.7 | 24.0±9.3 | 56.1±17.3 | 24.1±20.2 | 37.5±10.5 |
| | m-e | 0 | 97.7±.14.3 | - | 106.7±15.3 | - | 107.8±16.1 | - | 105.2±16.6 | - | 111.3±8.9 | - |
| | | 0.05 | 45.6±17.9 | 74.4±31.7 | 35.5±17.5 | 78.2±26.6 | 42.6±14.1 | 80.7±25.0 | 42.0±14.6 | 74.3±23.5 | 65.4±31.0 | 81.0±23.9 |
| | | 0.10 | 33.4±16.3 | 57.2±22.1 | 30.9±13.3 | 53.5±21.9 | 36.2±13.2 | 37.2±19.9 | 29.8±8.3 | 52.6±17.3 | 30.6±16.7 | 66.9±26.3 |
| | m | 0 | 59.1±12.6 | - | 73.6±14.2 | - | 59.7±9.3 | - | 80.4±16.3 | - | 102.6±15.1 | - |
| | | 0.10 | 29.5±9.2 | 50.1±16.3 | 37.8±13.9 | 32.9±21.7 | 30.2±12.1 | 51.3±14.1 | 33.9±12.3 | 57.9±22.2 | 42.3±22.9 | 57.3±21.3 |
| | | 0.15 | 20.7±8.3 | 43.9±18.8 | 25.8±11.3 | 30.9±20.0 | 20.2±9.3 | 42.5±17.2 | 21.9±7.9 | 51.3±21.9 | 21.6±13.0 | 33.9±17.0 |
| | m-r | 0 | 78.7±19.6 | - | 96.9±8.8 | - | 80.9±24.5 | - | 95.7±17.2 | - | 103.1±0.8 | - |
| | | 0.10 | 15.7±9.0 | 66.8±17.3 | 47.5±21.6 | 89.1±12.4 | 14.4±12.3 | 71.9±24.5 | 25.9±12.4 | 85.9±20.9 | 85.9±29.5 | 103.2±1.3 |
| | | 0.15 | 11.1±7.2 | 64.5±17.2 | 33.7±21.2 | 80.7±16.5 | 9.6±7.3 | 66.1±22.8 | 17.9±11.5 | 72.2±22.9 | 51.1±22.3 | 104.2±3.2 |
| | f-r | 0 | 89.7±23.9 | - | 102.1±2.1 | - | 78.8±27.2 | - | 106.7±0.1 | - | 107.5±3.3 | - |
| | | 0.10 | 17.4±7.3 | 73.1±29.0 | 54.3±25.0 | 103.3±4.2 | 19.4±7.7 | 67.2±30.3 | 34.9±10.2 | 96.0±11.4 | 69.2±19.8 | 101.6±7.5 |
| | | 0.15 | 11.7±6.9 | 64.9±25.6 | 31.1±11.1 | 97.3±19.9 | 9.4±5.2 | 64.2±27.8 | 22.9±10.2 | 83.2±22.6 | 52.5±26.3 | 99.5±10.8 |
| Walker2d | e | 0 | 111.6±0.6 | - | 108.8±1.9 | - | 110.7±0.5 | - | 109.6±0.5 | - | 104.8±12.5 | - |
| | | 0.10 | 77.9±37.6 | 110.3±2.0 | 97.6±21.9 | 94.3±20.3 | 72.9±39.4 | 109.2±1.5 | 93.3±27.2 | 109.1±4.0 | 95.4±19.7 | 97.8±20.2 |
| | | 0.15 | 28.2±32.4 | 104.2±13.5 | 78.9±33.2 | 83.4±23.3 | 9.2±13.6 | 107.5±5.2 | 30.5±32.5 | 94.5±18.1 | 81.6±26.4 | 84.5±26.4 |
| | m-e | 0 | 111.8±0.6 | - | 107.3±12.0 | - | 110.6±0.9 | - | 110.2±0.7 | - | 114.3±1.1 | - |
| | | 0.10 | 60.2±36.2 | 108.7±3.3 | 90.8±21.3 | 105.1±6.7 | 78.5±26.1 | 108.1±4.5 | 91.8±24.9 | 108.6±3.5 | 110.6±3.7 | 113.2±1.2 |
| | | 0.15 | 40.7±32.3 | 105.3±6.7 | 73.4±29.2 | 97.3±10.4 | 37.3±28.3 | 104.5±11.4 | 50.7±35.9 | 106.8±8.4 | 111.4±3.6 | 105.3±8.2 |
| | m | 0 | 77.3±15.3 | - | 82.3±3.1 | - | 83.6±10.1 | - | 86.4±1.2 | - | 101.5±0.8 | - |
| | | 0.10 | 67.2±22.0 | 74.4±18.3 | 76.8±15.8 | 89.3±8.1 | 81.5±11.3 | 84.9±7.7 | 77.7±20.4 | 85.2±7.3 | 98.6±7.5 | 93.3±11.0 |
| | | 0.15 | 48.7±28.6 | 65.8±20.7 | 64.4±22.9 | 77.2±9.2 | 62.3±29.8 | 80.6±10.2 | 53.6±28.3 | 78.3±14.4 | 82.3±18.3 | 90.0±14.2 |
| | m-r | 0 | 50.6±31.6 | - | 79.9±4.8 | - | 84.7±9.8 | - | 93.1±10.9 | - | 88.7±1.9 | - |
| | | 0.10 | 14.7±11.1 | 53.1±28.5 | 70.8±18.9 | 78.7±7.2 | 40.7±25.3 | 84.4±8.7 | 59.6±31.8 | 92.6±10.6 | 88.6±1.1 | 88.4±2.5 |
| | | 0.15 | 11.2±5.9 | 52.9±29.9 | 48.6±26.5 | 73.6±10.1 | 16.5±12.8 | 77.9±17.2 | 19.2±15.7 | 91.3±9.6 | 89.4±1.2 | 89.0±4.5 |
| | f-r | 0 | 82.2±23.3 | - | 92.8±2.2 | - | 96.7±1.7 | - | 99.2±1.4 | - | 104.3±0.7 | - |
| | | 0.10 | 38.5±28.5 | 85.6±19.7 | 89.2±6.9 | 91.3±3.2 | 69.9±28.3 | 96.3±2.2 | 79.2±29.5 | 98.7±2.8 | 99.2±7.4 | 101.5±6.7 |
| | | 0.15 | 19.3±18.3 | 77.1±25.0 | 68.6±28.1 | 89.1±6.1 | 23.3±22.1 | 92.2±5.5 | 18.5±23.4 | 97.6±4.7 | 71.2±33.9 | 98.5±11.8 |

Table 7: D4RL score of the baseline algorithms and the DMBP strengthened ones in Mujoco domain under uniformly distributed random noise (U-rand), maximum action-difference attack (MAD), and minimum Q-value attack (MinQ) on state observation.

| Env | Dataset/ Noise Scale | Noise Type | BCQ base | BCQ DMBP | CQL base | CQL DMBP | TD3+BC base | TD3+BC DMBP | Diffusion QL base | Diffusion QL DMBP | RORL base | RORL DMBP |
|---|---|---|---|---|---|---|---|---|---|---|---|---|
| HalfCheetah | e 0.05 | U-rand | 7.4±4.9 | 69.1±21.5 | 27.2±6.4 | 69.6±22.4 | 16.3±13.1 | 84.2±17.1 | 11.6±10.9 | 77.8±21.8 | 24.3±7.5 | 66.8±27.0 |
| | | MAD | 3.6±1.7 | 52.5±17.9 | 12.4±6.9 | 61.2±19.7 | 4.7±3.5 | 65.4±16.0 | 4.3±3.2 | 62.9±13.2 | 14.1±2.5 | 54.3±27.1 |
| | | MinQ | 12.8±9.3 | 51.8±23.9 | 19.4±11.3 | 60.4±19.4 | 18.0±4.2 | 88.2±11.3 | 8.0±6.7 | 71.1±15.2 | 9.3±8.8 | 71.0±29.1 |
| | m-e 0.05 | U-rand | 39.9±10.4 | 68.2±17.3 | 29.1±13.1 | 69.7±21.5 | 42.2±8.9 | 76.9±9.5 | 44.5±9.0 | 80.7±13.2 | 41.5±15.1 | 73.5±13.7 |
| | | MAD | 38.7±2.8 | 54.5±13.0 | 10.9±5.4 | 49.5±16.0 | 34.5±3.2 | 72.5±9.0 | 32.6±5.7 | 76.2±11.1 | 22.5±5.4 | 69.5±15.6 |
| | | MinQ | 36.3±6.1 | 51.7±12.9 | 26.9±12.5 | 58.5±24.0 | 30.8±7.3 | 78.3±12.6 | 45.0±2.0 | 71.5±10.2 | 55.8±13.2 | 79.5±4.5 |
| | m 0.05 | U-rand | 44.9±2.1 | 47.5±0.7 | 46.7±1.1 | 48.9±0.9 | 45.9±1.1 | 48.3±0.7 | 44.7±5.2 | 48.3±0.7 | 54.3±7.2 | 64.9±2.3 |
| | | MAD | 40.3±1.6 | 46.4±2.5 | 41.0±1.3 | 48.9±0.9 | 39.1±2.0 | 48.3±0.8 | 36.9±3.1 | 49.2±0.8 | 43.5±7.9 | 49.9±5.3 |
| | | MinQ | 34.3±9.8 | 46.1±2.7 | 45.6±0.9 | 49.0±1.1 | 42.2±2.0 | 48.3±0.8 | 43.3±0.8 | 43.2±1.0 | 51.6±5.4 | 55.3±5.0 |
| | m-r 0.10 | U-rand | 31.5±10.6 | 40.3±5.9 | 40.9±2.6 | 46.4±1.8 | 36.9±6.6 | 46.9±1.1 | 38.5±5.7 | 46.8±0.9 | 39.9±2.3 | 61.2±1.1 |
| | | MAD | 19.2±8.2 | 29.4±6.9 | 29.0±2.6 | 46.5±0.9 | 27.1±3.4 | 36.2±0.9 | 22.3±3.8 | 34.5±5.5 | 22.5±1.5 | 62.3±1.0 |
| | | MinQ | 5.1±5.2 | 36.7±8.8 | 39.2±0.8 | 46.2±1.1 | 36.7±6.8 | 44.8±1.1 | 37.0±4.8 | 38.6±1.1 | 34.0±1.4 | 63.2±2.3 |
| | f-r 0.10 | U-rand | 36.5±5.8 | 67.0±3.3 | 40.1±1.9 | 70.4±1.7 | 32.2±11.3 | 66.5±5.4 | 36.9±7.8 | 69.5±1.8 | 45.3±10.2 | 79.5±8.0 |
| | | MAD | 26.5±3.6 | 58.4±5.5 | 24.2±4.2 | 62.3±1.9 | 19.6±7.6 | 53.3±4.4 | 21.8±2.3 | 63.2±1.5 | 25.3±13.2 | 66.7±14.3 |
| | | MinQ | 13.8±9.2 | 46.1±11.9 | 36.2±1.5 | 53.4±3.7 | 20.4±12.8 | 67.2±1.5 | 35.6±1.4 | 53.1±1.8 | 53.5±16.9 | 72.4±6.2 |
| Hopper | e 0.05 | U-rand | 46.1±20.7 | 66.9±26.3 | 59.6±29.4 | 95.7±23.8 | 42.6±28.4 | 84.0±27.4 | 53.2±20.8 | 84.4±25.3 | 85.3±37.0 | 81.9±25.2 |
| | | MAD | 31.1±14.4 | 53.2±24.2 | 22.6±13.9 | 73.9±27.9 | 27.2±10.9 | 60.3±27.2 | 36.8±9.0 | 37.1±12.3 | 36.6±22.2 | 59.0±13.8 |
| | | MinQ | 47.4±18.9 | 62.5±27.9 | 32.7±13.5 | 58.7±17.9 | 45.3±27.5 | 95.7±27.6 | 66.7±33.6 | 59.2±23.9 | 79.8±32.7 | 59.4±22.1 |
| | m-e 0.05 | U-rand | 62.0±28.6 | 84.7±21.5 | 64.8±29.4 | 96.2±25.8 | 60.2±29.3 | 88.8±23.0 | 64.8±25.5 | 96.2±16.4 | 85.3±31.2 | 89.2±27.1 |
| | | MAD | 18.3±13.3 | 50.1±27.5 | 65.6±19.9 | 85.4±25.2 | 22.2±7.4 | 73.3±30.4 | 35.6±11.3 | 59.5±11.2 | 65.2±29.0 | 67.5±31.2 |
| | | MinQ | 28.5±18.1 | 43.2±13.9 | 74.1±28.2 | 81.2±29.4 | 88.4±31.5 | 101.1±21.7 | 38.3±25.6 | 69.9±32.3 | 93.3±27.9 | 85.9±28.2 |
| | m 0.05 | U-rand | 47.8±13.2 | 62.2±17.1 | 62.9±16.1 | 72.2±19.3 | 48.3±11.3 | 58.5±12.0 | 47.8±17.2 | 62.2±17.7 | 71.5±23.8 | 82.3±17.6 |
| | | MAD | 37.3±10.7 | 48.3±13.2 | 49.8±14.7 | 36.6±17.5 | 45.7±16.0 | 48.7±11.1 | 29.2±16.2 | 45.8±12.2 | 66.3±27.2 | 71.9±20.5 |
| | | MinQ | 27.9±10.8 | 45.2±10.5 | 59.9±18.7 | 87.0±27.9 | 49.7±18.2 | 61.3±13.6 | 59.2±18.6 | 75.4±12.9 | 63.9±21.7 | 77.5±18.2 |
| | m-r 0.10 | U-rand | 18.5±8.2 | 68.9±19.2 | 66.3±20.1 | 95.9±8.8 | 20.6±9.1 | 65.4±22.0 | 33.9±10.7 | 94.9±17.7 | 80.7±28.0 | 103.5±1.5 |
| | | MAD | 5.1±5.0 | 37.5±26.1 | 32.1±15.9 | 88.9±13.7 | 6.1±5.5 | 64.3±21.8 | 9.9±8.1 | 38.3±15.8 | 51.6±30.7 | 97.5±2.5 |
| | | MinQ | 5.3±5.4 | 18.3±18.4 | 84.6±14.1 | 87.5±6.6 | 11.8±7.6 | 80.5±18.1 | 51.2±25.1 | 62.5±27.3 | 98.3±6.2 | 103.2±2.4 |
| | f-r 0.10 | U-rand | 28.3±10.3 | 75.2±26.8 | 86.8±24.7 | 100.7±7.5 | 26.7±10.0 | 73.3±29.9 | 46.7±18.6 | 105.8±10.1 | 99.9±11.1 | 98.6±13.2 |
| | | MAD | 8.6±5.5 | 55.9±28.8 | 19.5±17.1 | 96.5±1.0 | 6.3±5.8 | 69.9±23.8 | 16.2±7.3 | 58.5±26.2 | 65.4±26.7 | 77.3±28.0 |
| | | MinQ | 14.7±8.1 | 17.0±18.1 | 99.1±1.8 | 102.3±4.6 | 6.8±6.0 | 77.9±29.6 | 66.0±24.2 | 80.2±22.6 | 82.5±20.9 | 91.7±16.3 |
| Walker2d | e 0.10 | U-rand | 102.1±1.8 | 110.4±0.8 | 106.1±9.9 | 106.0±7.4 | 106.1±2.9 | 110.0±0.5 | 107.2±1.0 | 109.4±0.5 | 95.1±15.7 | 97.2±9.5 |
| | | MAD | 50.5±43.7 | 70.5±13.3 | 64.1±27.0 | 97.6±16.1 | 19.9±22.7 | 69.7±17.5 | 36.6±35.5 | 88.2±24.8 | 61.9±29.2 | 83.8±19.9 |
| | | MinQ | 99.9±22.2 | 105.6±1.1 | 99.9±11.8 | 102.4±6.9 | 91.9±22.4 | 105.5±1.3 | 101.1±2.0 | 102.4±1.3 | 91.8±28.0 | 89.3±13.3 |
| | m-e 0.10 | U-rand | 91.9±26.5 | 110.7±1.1 | 102.7±11.2 | 108.7±7.5 | 90.7±22.8 | 110.1±1.0 | 103.3±13.7 | 109.5±0.8 | 108.3±3.2 | 106.2±4.5 |
| | | MAD | 38.3±33.7 | 98.5±7.5 | 73.2±34.7 | 102.5±16.1 | 22.8±21.0 | 78.5±5.4 | 37.3±34.1 | 99.5±11.3 | 88.6±21.7 | 101.9±6.0 |
| | | MinQ | 75.5±23.7 | 105.0±5.3 | 73.7±29.8 | 100.9±10.2 | 70.2±20.3 | 92.9±14.1 | 96.7±19.1 | 101.9±12.1 | 93.5±16.6 | 95.7±14.2 |
| | m 0.10 | U-rand | 77.8±16.2 | 76.1±15.7 | 82.6±5.5 | 81.5±3.7 | 82.6±13.7 | 84.9±4.2 | 85.5±5.9 | 85.3±5.5 | 99.7±3.8 | 101.0±0.6 |
| | | MAD | 38.6±27.4 | 74.7±16.8 | 59.3±29.5 | 80.5±5.6 | 41.1±34.3 | 74.1±8.1 | 32.2±29.5 | 61.5±13.7 | 77.3±26.6 | 79.2±21.3 |
| | | MinQ | 69.2±18.8 | 75.3±11.1 | 71.7±2.0 | 68.7±4.9 | 77.6±9.2 | 74.6±3.2 | 74.2±12.4 | 82.8±5.1 | 90.0±15.0 | 91.2±16.7 |
| | m-r 0.15 | U-rand | 17.3±12.2 | 54.9±25.7 | 69.2±20.9 | 78.1±9.2 | 51.2±28.3 | 83.6±14.8 | 64.2±27.8 | 91.1±12.1 | 89.9±1.1 | 88.7±2.1 |
| | | MAD | 6.6±3.3 | 43.4±29.8 | 19.7±14.7 | 78.4±8.8 | 8.8±4.4 | 70.8±19.1 | 7.2±2.3 | 66.1±24.2 | 81.9±11.5 | 90.5±3.5 |
| | | MinQ | 7.3±4.2 | 30.3±26.1 | 66.5±11.8 | 78.5±4.2 | 21.7±15.9 | 76.4±14.9 | 47.2±23.2 | 68.0±19.5 | 82.3±1.4 | 89.6±1.7 |
| | f-r 0.15 | U-rand | 40.9±30.6 | 83.4±21.4 | 88.7±7.7 | 91.9±2.3 | 60.2±28.1 | 96.2±2.4 | 85.7±21.1 | 98.2±3.8 | 103.2±3.3 | 98.7±9.0 |
| | | MAD | 5.8±4.5 | 66.1±26.3 | 36.6±29.7 | 86.3±9.0 | 7.6±1.8 | 76.3±2.9 | 5.9±1.8 | 63.9±10.9 | 75.3±21.3 | 72.9±20.6 |
| | | MinQ | 11.1±7.1 | 58.6±31.3 | 79.3±2.8 | 84.1±2.5 | 39.4±25.4 | 86.6±5.5 | 63.8±27.8 | 79.3±2.0 | 89.3±14.5 | 97.5±7.0 |

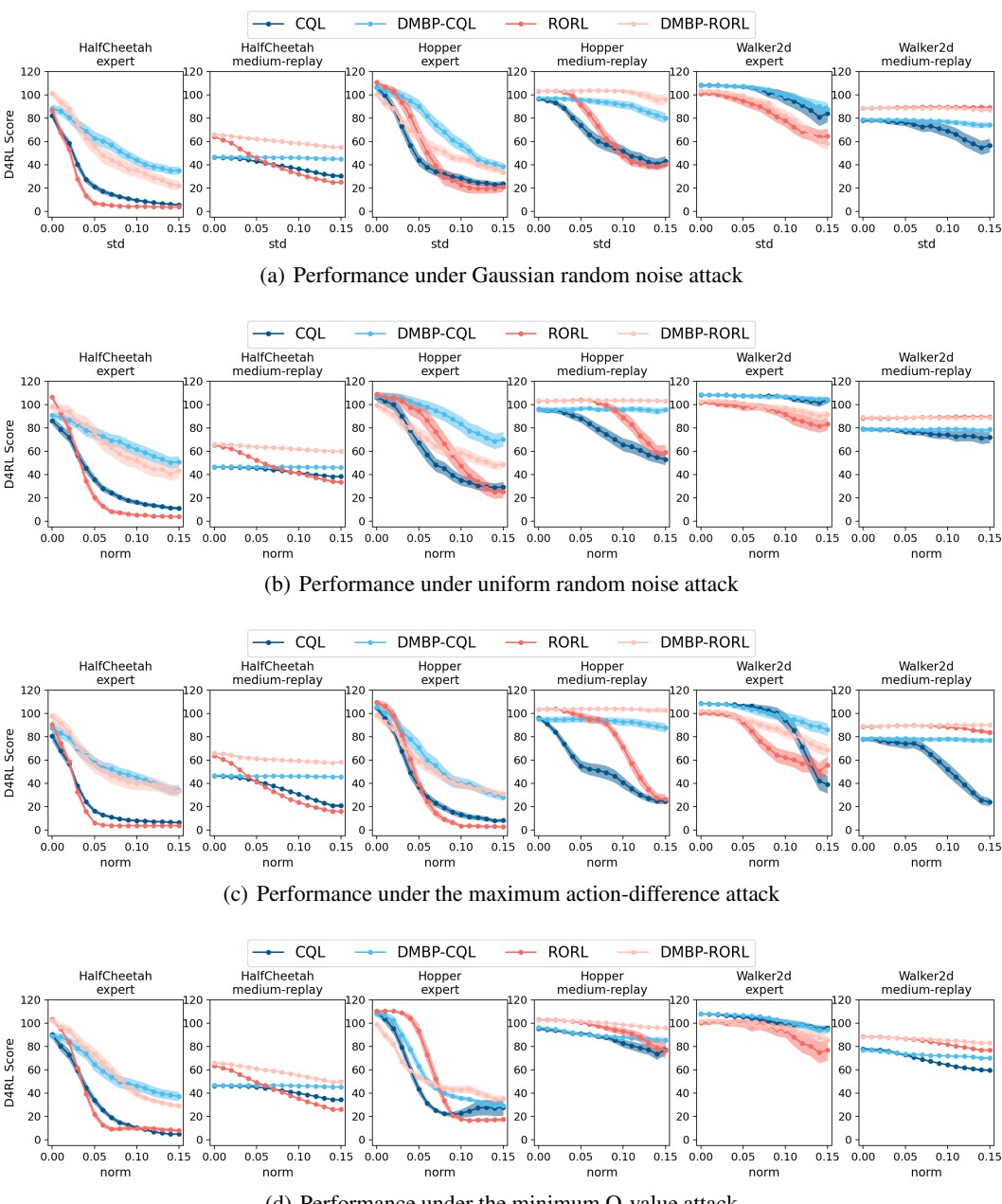

Figure 9: The performance of CQL, DMBP-CQL, RORL, and DMBP-RORL under different scales of (a) Gaussian random noise attack, (b) uniform random noise attack, (c) maximum action difference attack, and (d) minimum Q-value attack. The curves are smoothed with a window size of 3 and the shaded region represents half a standard deviation over 5 random checkpoints (20 tests for each checkpoint).

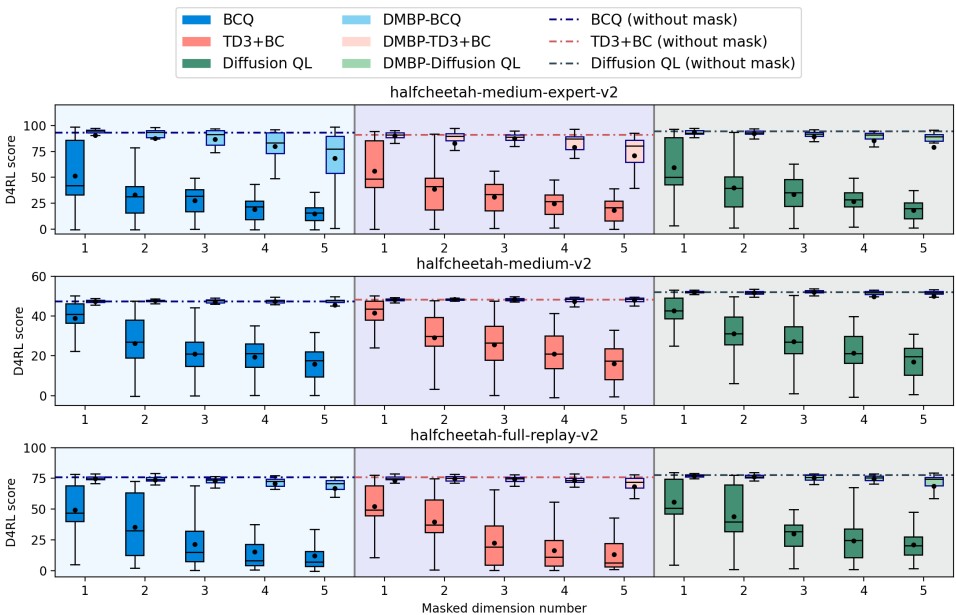

Figure 10: The performance of BCQ, DMBP-BCQ, TD3+BC, DMBP-TD3+BC, Diffusion QL, and DMBP-Diffusion QL (trained on halfcheetah medium-expert, medium, and full-replay datasets) with incomplete state observations that have 1-5 unobserved dimension(s). The dimension of state observation in halfcheetah is 17.

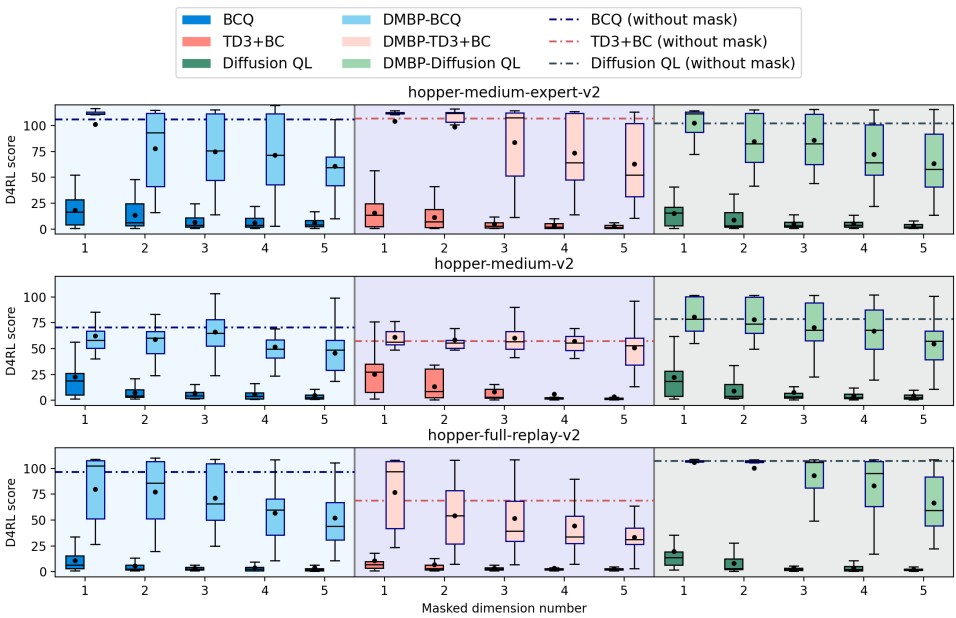

Figure 11: The performance of BCQ, DMBP-BCQ, TD3+BC, DMBP-TD3+BC, Diffusion QL, and DMBP-Diffusion QL (trained on hopper medium-expert, medium, and full-replay datasets) with incomplete state observations that have 1-5 unobserved dimension(s). The dimension of state observation in hopper is 11.

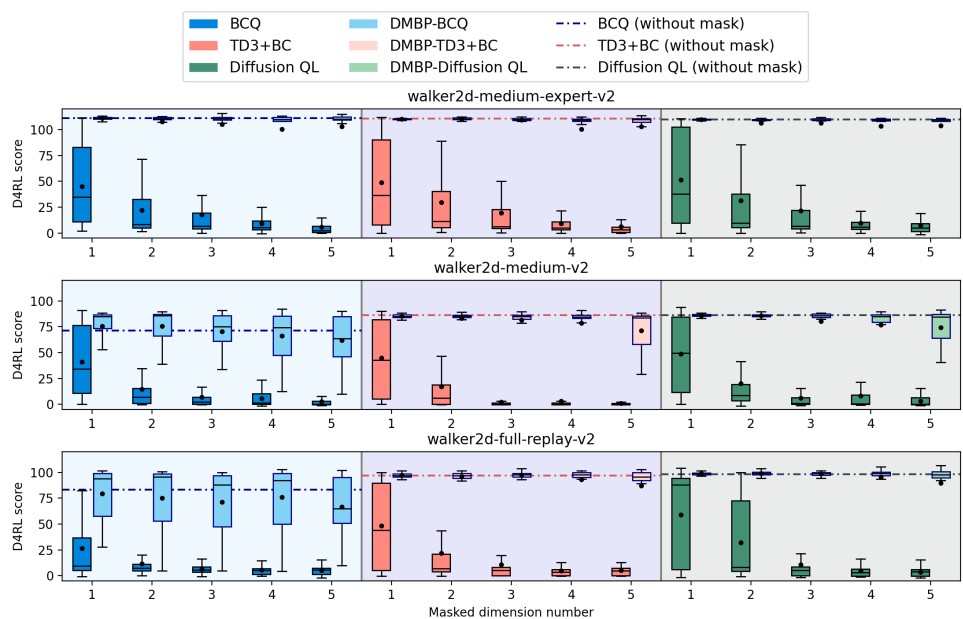

Figure 12: The performance of BCQ, DMBP-BCQ, TD3+BC, DMBP-TD3+BC, Diffusion QL, and DMBP-Diffusion QL (trained on walker2d medium-expert, medium, and full-replay datasets) with incomplete state observations that have 1-5 unobserved dimension(s). The dimension of state observation in walker2d is 17.

## D.3 Experimental results on Adroit

We further evaluate DMBP in the Adroit domain, where agents are required to control a 24-DoF robotic hand to manipulate a pen, a hammer, a door, and a ball. The action and state spaces in the Adroit domain have much higher dimensions than those in the Mujoco domain, and most of the baseline algorithms perform poorly on "human" and "cloned" datasets. Therefore, we only present the robustness evaluation results on the "expert" dataset as shown in Table 8. Again, DMBP significantly enhances the robustness of all baseline algorithms. Stronger robustness enhancement is observed for "policy regularization" offline RL algorithms (*i.e.*, *BCQ* and *Diffusion QL*), compared to "Q-value constrain" offline RL algorithms (*i.e.*, *CQL* and *RORL*).

For the robustness evaluation against in-complete state observations (cf. Figure 13), we mask more state dimensions (ranging from 2 to 20), and the evaluation results demonstrate that DMBP performs well even when roughly half of the state dimensions are masked, especially in "hammer" and "door" environments.

## D.4 Experimental results on Franka Kitchen

To demonstrate the efficacy of DMBP, we conduct experiments on the Franka Kitchen domain, where agents are required to control a 9-DoF Franka robot in a kitchen environment to finish multiple tasks on common household items. It is noticed that the Franka Kitchen domain has sparse reward environments, *i.e.*, agent scores 25 when it finishes a given task and scores 0 otherwise. Therefore, it is challenging to train an agent in such environments, and it becomes extremely difficult for an agent to obtain any rewards with perturbed state observations. We present the robustness evaluation results of DMBP against noised state observations in Table 9 and against incomplete state observations with unobserved dimensions in Figure 14. We note that DMBP significantly enhances the robustness of all baseline algorithms.

Table 8: D4RL score of the baseline algorithms and the DMBP strengthened ones in Adroit domain under Gaussian distributed random noise (G-rand), uniformly distributed random noise (U-rand), maximum action-difference attack (MAD), and minimum Q-value attack (MinQ) on state observations.

| Env/ dataset | Noise Scale | Noise Type | BCQ base | BCQ DMBP | CQL base | CQL DMBP | Diffusion QL base | Diffusion QL DMBP | RORL base | RORL DMBP |
|---|---|---|---|---|---|---|---|---|---|---|
| pen-expert-v1 | 0 | - | 121.2±38.5 | - | 104.5±66.5 | - | 143.4±25.4 | - | 126.3±55.5 | - |
| | 0.10 | G-rand | 114.3±48.3 | 111.5±41.2 | 69.7±45.4 | 89.7±55.3 | 131.0±37.6 | 141.1±20.5 | 121.1±31.0 | 124.8±43.7 |
| | | U-rand | 116.9±38.1 | 114.8±39.2 | 83.3±73.2 | 95.5±62.8 | 130.7±32.3 | 137.8±39.9 | 128.3±54.9 | 131.9±46.9 |
| | | MAD | 112.6±42.9 | 113.5±46.2 | 76.9±52.5 | 87.6±59.8 | 122.7±58.0 | 131.5±25.9 | 123.5±47.1 | 121.8±32.2 |
| | | MinQ | 118.7±42.2 | 121.1±49.7 | 83.5±66.3 | 76.1±66.9 | 129.0±43.9 | 142.6±23.7 | 114.9±48.7 | 126.4±41.6 |
| | 0.15 | G-rand | 108.4±50.9 | 114.1±43.2 | 51.3±42.5 | 71.6±65.9 | 128.3±43.9 | 143.1±39.5 | 102.6±56.1 | 124.0±48.5 |
| | | U-rand | 115.2±49.5 | 119.8±39.5 | 68.8±69.3 | 83.0±66.9 | 131.7±39.2 | 130.8±48.7 | 121.5±27.5 | 121.0±35.8 |
| | | MAD | 111.6±62.9 | 117.1±40.3 | 58.3±59.8 | 65.6±55.2 | 116.6±54.9 | 129.4±32.8 | 103.0±57.1 | 110.2±53.5 |
| | | MinQ | 108.4±42.8 | 113.1±38.3 | 69.3±55.9 | 97.5±59.5 | 119.6±49.5 | 135.4±45.6 | 111.3±48.8 | 122.4±38.2 |
| hammer-expert-v1 | 0 | - | 108.5±1.2 | - | 85.4±31.5 | - | 105.7±4.8 | - | 111.7±53.1 | - |
| | 0.10 | G-rand | 29.7±48.7 | 109.1±1.3 | 5.7±1.0 | 48.3±15.5 | 68.6±49.8 | 103.6±16.7 | 11.7±0.2 | 38.3±56.5 |
| | | U-rand | 86.9±28.2 | 107.3±13.0 | 29.6±1.0 | 76.0±21.7 | 84.9±47.5 | 107.3±1.5 | 33.2±38.4 | 60.5±40.2 |
| | | MAD | 5.8±11.9 | 84.9±31.5 | -0.1±0.3 | 18.5±12.5 | 16.4±18.4 | 78.5±31.0 | 0.0±0.1 | 18.9±11.9 |
| | | MinQ | 83.3±37.9 | 109.9±1.2 | 15.5±23.0 | 52.2±26.8 | 33.3±32.2 | 103.5±11.7 | 11.2±1.7 | 46.9±37.3 |
| | 0.15 | G-rand | 0.4±0.6 | 101.8±5.4 | 0.0±0.3 | 21.6±0.7 | 33.3±46.5 | 100.1±26.2 | 0.1±0.2 | 11.7±0.4 |
| | | U-rand | 51.1±56.6 | 109.0±5.9 | 10.6±9.2 | 55.9±23.9 | 79.0±56.5 | 106.1±7.2 | 4.5±6.1 | 40.3±35.7 |
| | | MAD | 0.1±0.0 | 69.0±22.3 | -0.1±0.0 | 11.9±13.9 | 4.0±11.3 | 66.2±34.6 | -0.1±0.0 | 9.2±5.2 |
| | | MinQ | 26.4±47.3 | 99.3±1.2 | 0.9±1.2 | 31.3±22.5 | 0.5±0.9 | 99.2±32.9 | 0.1±0.1 | 22.2±13.3 |
| door-expert-v1 | 0 | - | 104.7±7.7 | - | 106.6±0.9 | - | 103.6±7.3 | - | 117.8±4.6 | - |
| | 0.10 | G-rand | 88.2±33.2 | 83.5±36.0 | 88.3±23.6 | 103.2±9.8 | 95.6±21.3 | 103.5±9.2 | 65.3±48.2 | 95.3±27.4 |
| | | U-rand | 104.4±5.3 | 102.9±10.6 | 103.0±3.7 | 103.6±14.3 | 101.5±18.5 | 105.1±1.4 | 86.5±40.4 | 98.5±26.7 |
| | | MAD | 97.3±19.6 | 103.2±6.9 | 31.0±21.7 | 74.5±26.6 | 92.9±35.8 | 103.7±7.8 | 30.4±23.8 | 86.6±36.5 |
| | | MinQ | 95.6±25.0 | 99.3±19.9 | 53.5±41.7 | 105.1±3.1 | 100.6±13.8 | 103.2±5.1 | 86.1±35.8 | 102.0±14.6 |
| | 0.15 | G-rand | 44.1±48.6 | 82.5±19.2 | 25.5±34.2 | 102.9±8.6 | 53.2±46.1 | 101.5±16.0 | 14.0±31.1 | 95.7±25.2 |
| | | U-rand | 95.9±24.7 | 100.9±9.8 | 89.2±26.2 | 98.7±20.4 | 93.7±24.0 | 101.1±14.0 | 73.7±42.2 | 95.9±18.1 |
| | | MAD | 37.7±37.7 | 84.2±15.8 | 0.2±0.4 | 62.8±38.5 | 54.8±35.8 | 81.9±27.8 | 7.4±11.7 | 73.7±27.7 |
| | | MinQ | 74.3±44.5 | 94.8±27.2 | 0.1±0.1 | 97.3±17.9 | 86.1±25.5 | 103.7±3.2 | 42.7±38.3 | 100.5±14.9 |
| relocate-expert-v1 | 0 | - | 66.2±30.8 | - | 76.2±36.6 | - | 101.2±12.7 | - | 43.3±20.5 | - |
| | 0.10 | G-rand | 2.0±4.9 | 42.0±35.5 | 0.9±2.9 | 34.6±31.7 | 15.3±25.1 | 96.6±24.3 | 1.1±0.1 | 27.3±11.5 |
| | | U-rand | 50.9±37.8 | 65.8±39.0 | 2.3±3.9 | 41.4±24.1 | 69.8±42.9 | 98.0±13.6 | 12.1±2.1 | 35.7±31.7 |
| | | MAD | 0.1±0.2 | 34.7±25.8 | 0.2±0.2 | 24.2±21.0 | 0.1±0.5 | 66.8±30.7 | 0.0±0.1 | 15.8±21.0 |
| | | MinQ | 57.7±35.3 | 52.9±29.8 | 0.2±0.3 | 37.5±25.8 | 3.4±4.2 | 70.6±37.1 | 0.9±0.8 | 34.2±22.1 |
| | 0.15 | G-rand | 0.2±0.4 | 48.3±33.7 | 0.0±0.1 | 34.2±31.2 | 1.3±3.5 | 87.2±20.5 | -0.1±0.0 | 22.0±15.5 |
| | | U-rand | 8.1±11.3 | 47.4±25.5 | 0.8±1.9 | 36.1±30.0 | 32.5±22.5 | 93.4±16.7 | 0.1±0.1 | 30.3±22.7 |
| | | MAD | 0.0±0.1 | 33.6±31.5 | 0.0±0.0 | 28.2±21.9 | 0.0±0.3 | 67.1±32.5 | -0.1±0.1 | 16.3±18.5 |
| | | MinQ | 17.5±17.2 | 42.2±23.5 | 0.1±0.1 | 36.6±25.6 | 0.7±0.4 | 50.2±31.6 | -0.1±0.0 | 9.4±5.4 |

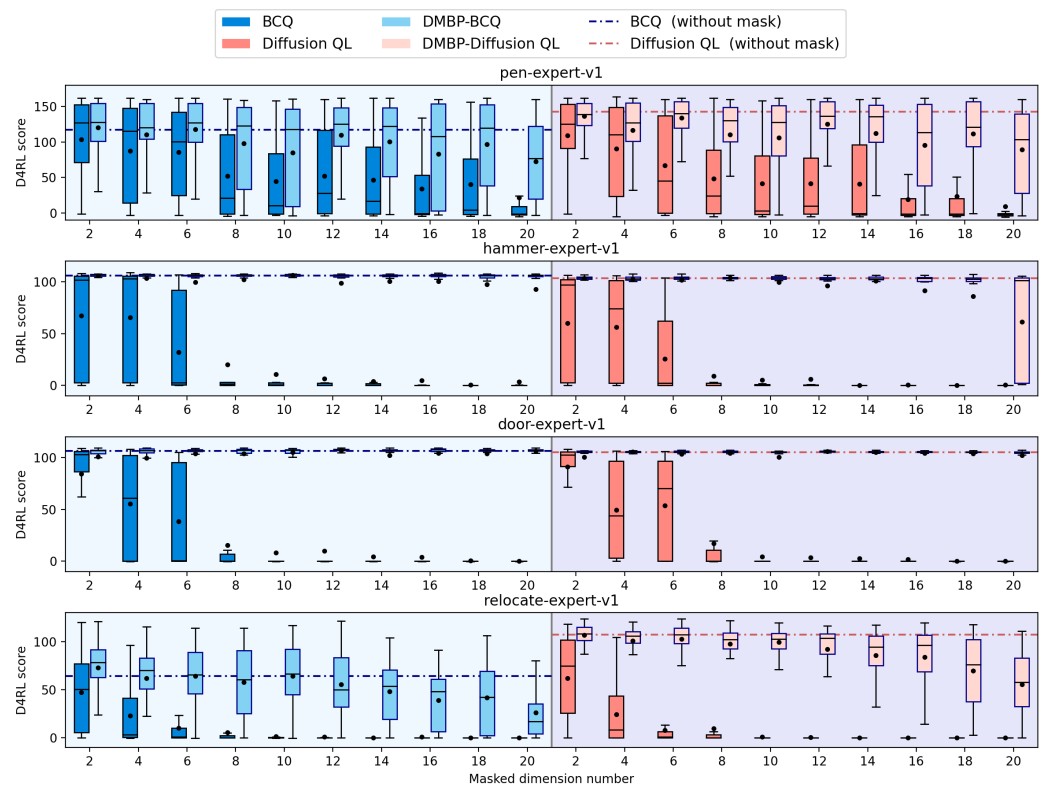

Figure 13: The performance of BCQ, DMBP-BCQ, Diffusion QL, and DMBP-Diffusion QL in Adroit domain with incomplete state observations that have 2-20 unobserved dimensions. (The dimension of state observation is 45 in pen, 46 in hammer, and 39 in both door and relocate.)

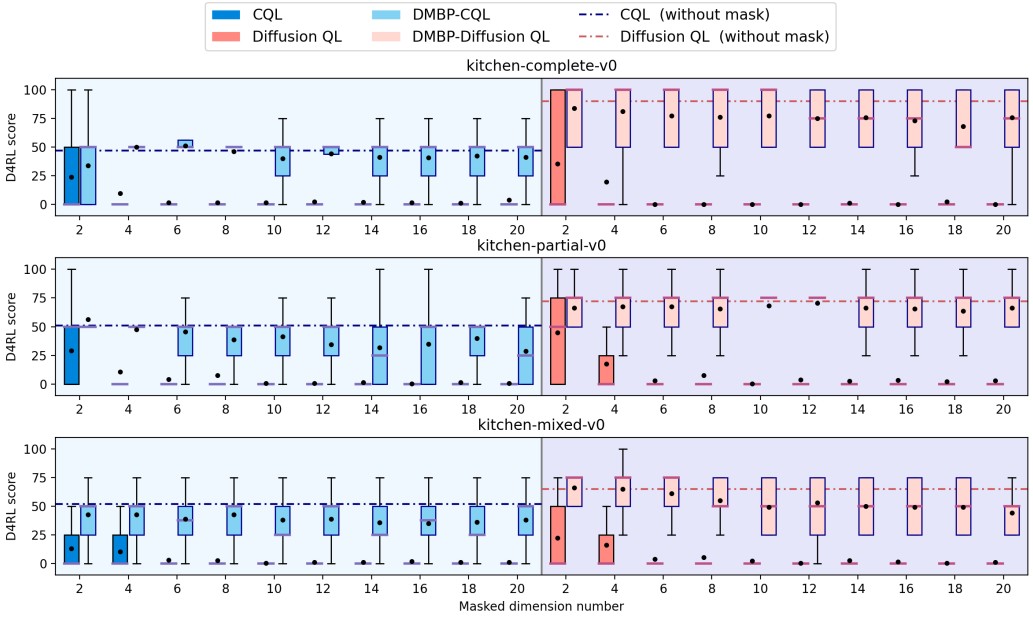

Figure 14: The performance of CQL, DMBP-CQL, Diffusion QL, and DMBP-Diffusion QL in Franka Kitchen domain with incomplete state observations that have 2-20 unobserved dimensions. (The dimension of state observation in kitchen is 59.)

Table 9: D4RL score of the baseline algorithms and the DMBP strengthened ones in Franka Kitchen domain under Gaussian distributed random noise (G-rand), uniformly distributed random noise (U-rand), maximum action-difference attack (MAD), and minimum Q-value attack (MinQ) on state observation.

| Env/dataset | Noise Scale | Noise Type | BCQ base | BCQ DMBP | CQL base | CQL DMBP | Diffusion QL base | Diffusion QL DMBP | RORL base | RORL DMBP |
|---|---|---|---|---|---|---|---|---|---|---|
| kitchen-complete-v0 | 0 | - | 7.5±15.9 | - | 47.3±32.4 | - | 84.5±27.7 | - | 22.8±26.8 | - |
| | 0.05 | G-rand | 0.8±4.3 | 5.0±12.9 | 0.0±0.0 | 34.5±25.6 | 0.8±4.3 | 65.8±22.4 | 0.0±0.0 | 12.5±17.8 |
| | | U-rand | 0.8±4.3 | 4.5±9.6 | 0.8±4.3 | 31.8±24.0 | 0.8±4.3 | 60.0±29.6 | 0.0±0.0 | 13.0±18.1 |
| | | MAD | 0.8±4.3 | 12.5±17.8 | 0.0±0.0 | 10.8±18.4 | 2.5±7.5 | 42.0±20.0 | 0.0±0.0 | 6.5±14.8 |
| | | MinQ | 0.0±0.0 | 6.3±12.8 | 0.8±4.3 | 17.3±20.2 | 0.0±0.0 | 56.8±28.0 | 0.0±0.0 | 9.8±17.9 |
| | 0.10 | G-rand | 0.3±2.5 | 1.3±5.4 | 0.0±0.0 | 32.5±24.4 | 0.0±0.0 | 61.0±30.5 | 0.0±0.0 | 6.0±14.2 |
| | | U-rand | 0.8±4.3 | 3.3±8.4 | 0.0±0.0 | 23.5±27.1 | 0.0±0.0 | 53.0±25.4 | 0.0±0.0 | 8.0±16.4 |
| | | MAD | 0.0±0.0 | 4.0±9.2 | 0.0±0.0 | 9.8±17.8 | 0.0±0.0 | 34.3±25.9 | 0.0±0.0 | 3.5±8.7 |
| | | MinQ | 0.0±0.0 | 5.8±10.6 | 0.0±0.0 | 23.3±27.0 | 0.0±0.0 | 47.8±22.4 | 0.0±0.0 | 6.5±14.6 |
| kitchen-partial-v0 | 0 | - | 29.3±18.3 | - | 51.3±34.5 | - | 71.8±23.2 | - | 30.3±23.0 | - |
| | 0.05 | G-rand | 6.3±11.7 | 11.8±19.0 | 0.0±0.0 | 32.5±24.4 | 12.5±17.8 | 65.8±22.4 | 0.0±0.0 | 18.8±20.9 |
| | | U-rand | 3.3±8.4 | 10.0±18.1 | 0.8±4.3 | 28.5±22.7 | 22.5±26.3 | 60.0±29.6 | 1.5±5.8 | 21.5±25.5 |
| | | MAD | 6.5±14.8 | 5.3±13.0 | 0.0±0.0 | 22.5±26.3 | 5.0±12.7 | 48.3±22.7 | 1.3±5.4 | 13.3±18.1 |
| | | MinQ | 5.0±12.9 | 5.0±12.7 | 3.3±8.4 | 29.8±22.9 | 4.3±12.1 | 40.0±28.6 | 0.0±0.0 | 15.3±19.5 |
| | 0.10 | G-rand | 0.8±4.3 | 7.3±11.5 | 0.0±0.0 | 26.8±22.3 | 7.3±15.8 | 64.3±21.3 | 0.0±0.0 | 16.0±19.8 |
| | | U-rand | 2.3±7.2 | 7.5±15.9 | 0.0±0.0 | 23.3±27.1 | 4.3±12.3 | 55.8±28.5 | 0.5±3.5 | 17.3±20.2 |
| | | MAD | 2.5±7.5 | 4.3±9.4 | 0.0±0.0 | 14.3±18.8 | 0.8±4.3 | 41.3±19.6 | 0.0±0.0 | 9.8±17.9 |
| | | MinQ | 5.0±10.0 | 10.8±18.4 | 0.0±0.0 | 12.5±20.0 | 3.3±8.4 | 39.5±18.1 | 0.0±0.0 | 12.0±19.8 |
| kitchen-mixed-v0 | 0 | - | 18.3±23.2 | - | 51.3±34.5 | - | 65.3±21.8 | - | 33.5±24.9 | - |
| | 0.05 | G-rand | 7.3±11.3 | 16.3±20.0 | 0.0±0.0 | 30.8±23.3 | 11.3±18.8 | 49.3±22.8 | 0.8±4.3 | 11.5±19.1 |
| | | U-rand | 9.3±17.6 | 18.8±20.9 | 0.0±0.0 | 32.8±24.6 | 22.5±26.2 | 55.0±26.9 | 0.0±0.0 | 12.3±19.8 |
| | | MAD | 5.8±14.0 | 15.0±19.2 | 0.0±0.0 | 26.3±22.0 | 13.3±18.2 | 42.3±20.1 | 2.3±7.2 | 13.0±18.1 |
| | | MinQ | 0.8±4.3 | 15.8±19.7 | 0.0±0.0 | 34.3±22.6 | 9.3±17.6 | 40.0±28.6 | 1.5±5.8 | 15.5±19.6 |
| | 0.10 | G-rand | 1.3±5.4 | 14.5±18.9 | 1.8±6.4 | 28.3±22.5 | 0.0±0.0 | 50.0±33.7 | 0.0±0.0 | 9.8±16.1 |
| | | U-rand | 5.0±10.0 | 16.8±20.0 | 1.5±5.8 | 30.3±22.9 | 1.5±5.8 | 50.8±34.3 | 0.0±0.0 | 11.5±18.8 |
| | | MAD | 0.0±0.0 | 12.5±20.2 | 0.0±0.0 | 17.8±20.3 | 2.5±7.5 | 40.0±28.6 | 0.8±4.3 | 12.8±17.8 |
| | | MinQ | 0.8±4.3 | 13.3±18.1 | 0.3±2.5 | 25.8±21.9 | 1.8±6.4 | 40.0±28.8 | 0.3±2.5 | 12.3±19.8 |

## D.5 COMPUTATIONAL COST

In Table 10, we compare the computational cost of DMBP for multiple baseline algorithms on a single machine with one GPU (NVIDIA RTX4090 24GB) and one CPU (AMD Ryzen 9 7950X). For each algorithm, we perform the training on halfcheetah-expert-v2 dataset, and measure the average epoch time (1000 gradient steps for each epoch) and the GPU memory usage over 10 epochs. Although the training of DMBP requires more computing time per epoch, DMBP expedites the convergence (in 200-300 epochs) of the baseline algorithms (which usually converge in 1000-3000 epochs), *i.e.*, the total training time of DMBP is similar to that of the baseline algorithms in comparison.

Table 10: Computational cost comparison

| Algorithms | Runtime ($s$/epoch) | GPU Memory (GB) |
|---|---|---|
| BCQ | 4.33 | 1.36 |
| CQL | 19.58 | 1.33 |
| TD3+BC | 2.82 | 1.29 |
| Diffusion QL | 12.05 | 1.69 |
| RORL | 22.63 | 1.44 |
| DMBP | 194.08 | 2.22 |

Table 11: Decision time consumption comparison (1000 timesteps)

| Decision algorithms | Runtime($s$) - | Decision algorithms | Runtime($s$) ($k=1$) | Runtime($s$) ($k=5$) |
|---|---|---|---|---|
| BCQ | 0.543 | DMBP-BCQ | 2.411 | 9.980 |
| CQL | 0.491 | DMBP-CQL | 2.323 | 9.776 |
| TD3+BC | 0.365 | DMBP-TD3+BC | 2.215 | 9.698 |
| Diffusion QL | 3.388 | DMBP-Diffusion QL | 4.786 | 12.241 |
| RORL | 0.525 | DMBP-RORL | 2.439 | 10.143 |

We also perform experiments to measure the decision time consumption for baseline RL algorithms and the DMBP improved ones (see Table 11). We compare the total runtime over 1000 RL timesteps of the baseline algorithms (trained on halfcheetah-expert-v2 dataset) with and without the DMBP denoising framework. For the start diffusion timestep of DMBP in denoising tasks, we select two values of $k = 1$ and $k = 5$. DMBP consumes more decision time within an acceptable range.

To expedite the training process of DMBP, one can optimize the training schedule by adjusting the number of diffusion steps (Chung et al., 2022; Franzese et al., 2022) and the design of noise scale (Nichol & Dhariwal, 2021) through learning, which have been proven effective in speeding up the convergence rate. Analogously, to expedite the testing process of DMBP, one can employ implicit sampling techniques, such as Song et al. (2020a); Zhang et al. (2022a), which aim to reduce the number of reverse denoising steps required during the testing phase. The proposed methods will be subject to verification in our future work.

### D.6    EXPERIMENTAL RESULT ANALYSIS

In our experiments, we find that DMBP achieves better robustness enhancement performance when it is combined with offline RL algorithms based on *policy regularization* (*BCQ, TD3BC, Diffusion QL*), as opposed to those based on *conservative Q estimation* (*CQL, RORL*). Among the baseline algorithms we evaluate, *Diffusion QL* demonstrates the best performance when paired with DMBP, while the least improvement exhibits for *RORL*. For the purpose of understanding the underlying mechanisms, we employ t-SNE (Van der Maaten & Hinton, 2008; Chan et al., 2018) to visualize the distributions of states and actions encountered during the training and testing processes of these offline RL algorithms.

As shown in Figure 15, we train various baseline offline RL algorithms using the halfcheetah-expert-v2 dataset. The visualization depicts the distributions of states and actions in the dataset, as well as during the testing process without any perturbations, across 20 epochs (each epoch consisting of 1000 RL timesteps). The test results indicate that *policy regularization* based algorithms tend to replicate the policies observed in the dataset during the testing process. They also exhibit a preference for accessing states that are present in the dataset. This phenomenon is particularly evident in the case of *Diffusion QL*, where the training and testing sets largely overlap. For *conservative Q* based algorithms, on the other hand, the overlap between the training and testing sets is less significant (particularly for *RORL*). By the nature of the Diffusion Models, DMBP has a tendency to recover states that frequently appear in the dataset. This explains why DMBP exhibits better robustness enhancement performance when paired with *policy regularization* based offline RL algorithms.

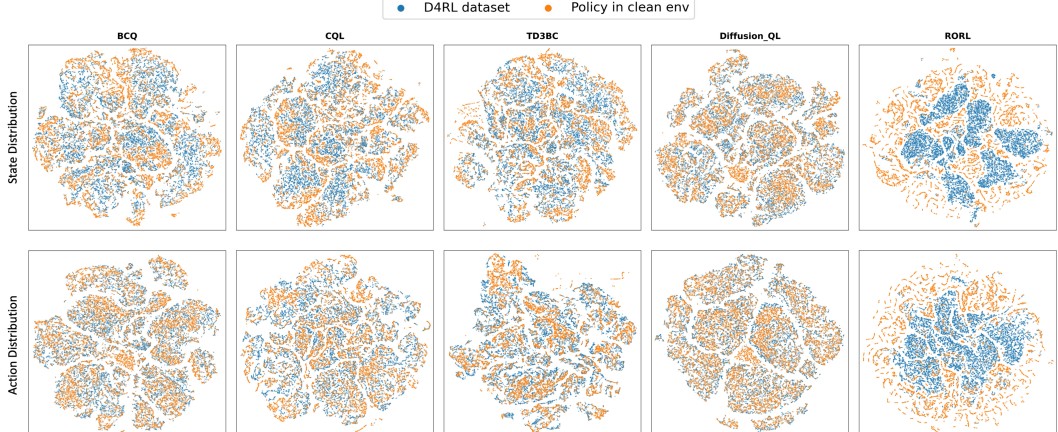

Figure 15: Visualization of the state and action distributions of the training datasets (halfcheetah-expert-v2), and the generated data during the testing process in the clean environment for various baseline offline RL algorithms (BCQ, CQL, TD3+BC, Diffusion QL, RORL).

We note that DMBP demonstrates a certain level of generalization ability for unseen states and actions in the dataset. We visualize the distributions of states and actions for the agent in the clean environments, the agent in the noised environments with the assistance of DMBP, and the agent in the noised environments without the assistance of DMBP (see Figure 16).[1] There is a significant overlap between the state and action distributions obtained by the agent in the clean environment (the green scatter plot) and those from the agent assisted by DMBP in the noisy environment (the red scatter plot). This indicates that DMBP is able to accurately infer the original states during testing, even in cases where some states are not observed in the training set. Without the assistance of DMBP, the distribution of actions taken by the agent in the noisy environment, as well as the distribution of states encountered during the process (the purple scatter plot), are completely different from those observed in the clean environment.

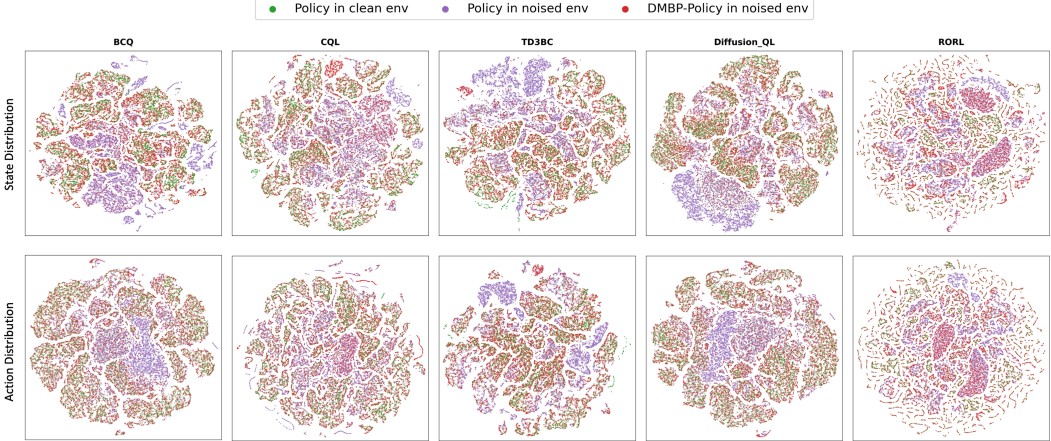

Figure 16: Visualization of the state and action distributions for the agent in the clean environments, the agent in the noised environments with the assistance of DMBP, and the agent in the noised environments without the assistance of DMBP. The noise is set to be Gaussian distributed with a std of 0.03. The baseline RL algorithms (BCQ, CQL, TD3+BC, Diffusion QL, RORL) and DMBP are all trained on halfcheetah-expert-v2.

## E    LIMITATIONS

One limitation of our present work is that the noise scale related hyperparameter (*i.e.*, the diffusion start timestep $k$ in Eq. 2) is manually defined when DMBP is deployed for online tests in environments with different scales of noises. In future work, we will consider self-adaptive diffusion start timestep to make DMBP more adaptive for environments with varying noise scales.

Another limitation of this work is the lack of validation of DMBP's performance with normalized state observations (as in Zhang et al. (2020); Yang et al. (2022)). Without normalizing the state observation, adding noise of the same scale may introduce a bias in the corruption's effect on different dimensions. In future work, we will validate and optimize the performance of DMBP with normalized state observations.

---

[1] Note that the recorded states during the testing process in the noised environment are the original states other than the noised ones.

