# OpenReview forum: "DMBP: Diffusion model-based predictor for robust offline reinforcement learning against state observation perturbations"
_ICLR.cc/2024/Conference — ICLR 2024 poster_

### Official Review · Reviewer_As3y · 2023-10-25

**Soundness:** 4 excellent
**Presentation:** 4 excellent
**Contribution:** 3 good
**Rating:** 8
**Confidence:** 3

**Summary:**

The paper presents DMBP, a novel framework for robust offline RL. DMBP addresses the challenge of handling state observation perturbations by employing conditional diffusion models to recover the true states from perturbed observations. Additionally, it proposes a non-Markovian training objective to reduce the cumulative errors by minimizing the sum entropy of denoised states along the RL trajectory. Experimental results show that DMBP significantly enhances the robustness of current offline RL algorithms, effectively addressing various perturbations and incomplete state observations.

**Strengths:**

* Overall, this paper is well-organized and easy to follow.

* The introduction of the diffusion model as a means to recover true states from perturbed observations, addressing the challenge of observation perturbation, is both novel and sound. Besides, the inclusion of the regularization of sum entropy for denoised states across the RL trajectory is highly meaningful in the context of sequential decision-making, effectively mitigating error accumulation.

* The proposed method can be applied to many model-free offline RL algorithms and significantly improves over prior works in terms of observation robustness and masked observations.

I believe this paper represents a significant contribution to the field and will have a substantial impact on the research community.

**Weaknesses:**

* The literature review section lacks a discussion of another taxonomy: training-time and testing-time robustness. While this paper and many others focus on offline RL for testing-time robustness, there is another group of offline RL works that investigate training-time robustness [1] [2] [3]. These works involve corrupting the offline dataset and evaluating it in a clean environment. It would be beneficial to include the training-time works for a more comprehensive literature review.

* It is not clear whether the states in this paper are normalized. Normalizing the observations would ensure fair observation corruption and may potentially impact the recovery ability of the diffusion model.

* Currently, it seems that the diffusion model is trained using complete observations even in the experiments with incomplete observations. I am curious to know if the diffusion model can handle training with incomplete observations as well.

* Additionally, the authors are suggested to report the training time and the inference time of the proposed method.

* Typo: page 2 'A diagram of the proposed approach is shown in the fight subplot of Figure 1' --> 'A diagram of the proposed approach is shown in the right subplot of Figure 1'.

[1] Wu F, Li L, Xu C, et al. Copa: Certifying robust policies for offline reinforcement learning against poisoning attacks[J]. arXiv preprint arXiv:2203.08398, 2022.

[2] Zhang X, Chen Y, Zhu X, et al. Corruption-robust offline reinforcement learning[C]//International Conference on Artificial Intelligence and Statistics. PMLR, 2022: 5757-5773.

[3] Ye C, Yang R, Gu Q, Zhang T. Corruption-Robust Offline Reinforcement Learning with General Function Approximation. arXiv:2310.14550.

**Questions:**

My questions are listed in the "Weakness" part:

* The authors are suggested to provide a more comprehensive literature review.

* It is necessary for the authors to clarify whether the observations are normalized. If not, it would be beneficial to include a comparison and provide further clarification.

* The reviewer is interested in knowing if the diffusion model can also handle training with incomplete observations.

* The authors are recommended to include information about the training time and inference time of the proposed method.

* Fix the typo.

---

> ### Author Response · Authors · 2023-11-15
> **Response**
>
> We acknowledge Reviewer As3y for appreciating the significance and novelty of our work and providing valuable suggestions. Below, we respond to the reviewer's questions point by point.
>
> 1. Thanks for pointing this out. In the Introduction, we added the review on training-time robustness and testing-time robustness as follows:
>
> Robust RL can be categorized into two taxonomies: training-time and testing-time
> robustness. Training-time robust RL involves perturbations during the training process, while
> evaluating the agent in a clean environment (Zhang et al., 2022; Ye et al., 2023). Conversely, testing-time robust RL focuses on training the agent with unperturbed datasets or environments and then
> testing its performance in the presence of disturbances (Yang et al., 2022; Panaganti et al., 2022). Our
> work primarily aims at enhancing the testing-time robustness of existing offline RL algorithms.
>
> 2. We do not normalize the observations in our experiments.
> In most experimental settings for testing the robustness of RL algorithms against uncertain observations, perturbed observations are selected within an $\ell_{\infty}$ ball, or the noise is added randomly with a specific scale. This approach ensures that the noise scale for each dimension of the state is the same.
> However, if we choose to normalize the states perturbed by the aforementioned noises, the normalized states would have different noise scales for different dimensions, since the distribution of each state dimension is distinct.
> Therefore, normalization of the perturbed states can complicate the denoising process for DMBP.
> We have added a paragraph in Appendix B.2 to clarify this point.
>
> 3. To our knowledge, DMBP does not possess the capability to handle training with incomplete observations. The primary objective of DMBP is to accurately recover the actual states for the agent to make decisions. However, it lacks the ability to reason about the unknown dimensions of states in offline datasets. Consequently, DMBP may require the baseline agent to be trained with unobserved dimensions in such scenarios.
> Unfortunately, due to the time constraints, we were unable to validate this approach within the revision period. We firmly believe that this avenue holds promising value for further exploration and research.
>
> 4. We have included a new section (Appendix D.5), which provides the computing costs associated with DMBP (including training and testing costs).
>
> 5. We apologize for the typos we made. We have carefully checked entire paper to correct all typos.
>
> [1] Xuezhou Zhang, Yiding Chen, Xiaojin Zhu, and Wen Sun. Corruption-robust offline reinforcement learning. In International Conference on Artificial Intelligence and Statistics, pp. 5757–5773. PMLR, 2022.
>
> [2] Chenlu Ye, Rui Yang, Quanquan Gu, and Tong Zhang. Corruption-robust offline reinforcement learning with general function approximation. arXiv preprint arXiv:2310.14550, 2023.
>
> [3] Rui Yang, Chenjia Bai, Xiaoteng Ma, Zhaoran Wang, Chongjie Zhang, and Lei Han. RORL: Robust
> offline reinforcement learning via conservative smoothing. In Advances in Neural Information
> Processing Systems, 2022.
>
> [4] Kishan Panaganti, Zaiyan Xu, Dileep Kalathil, and Mohammad Ghavamzadeh. Robust reinforcement
> learning using offline data. arXiv preprint arXiv:2208.05129, 2022.

---

> > ### Comment · Reviewer_As3y · 2023-11-21
> > **Thanks for your response**
> >
> > Thank you for the reply. I still have concerns regarding the lack of input normalization in the given setting. Normalization has been adopted in prior attack works [1][2] in the literature of robust RL. In the absence of normalization, noise of the same scale will have a minimal impact on dimensions with large scale values, but a significant impact on dimensions with small scale values. The corruption is biased for different dimensions. Although this issue may not be crucial considering the overall contribution of this work, I believe it can be addressed by (1) emphasizing the unnormalized setting in the main paper and mentioning it in the limitation section, and (2) providing some results in the normalized setting if possible.
> >
> > Another concern I have is the considerable computational cost of DMBP compared to other algorithms. I recommend that the authors include a discussion on potential future strategies to mitigate this issue and emphasize it as a limitation in the corresponding section.
> >
> > [1] Zhang H, et al. Robust deep reinforcement learning against adversarial perturbations on state observations[J]. Advances in Neural Information Processing Systems, 2020.
> >
> > [2] Rui Yang, et al. RORL: Robust offline reinforcement learning via conservative smoothing. In Advances in Neural Information Processing Systems, 2022.

---

> ### Author Response · Authors · 2023-11-21
> **Thanks for your suggestions**
>
> Thank you for the detailed description of your concerns.
>
> > I still have concerns regarding the lack of input normalization in the given setting. Normalization has been adopted in prior attack works [1][2] in the literature of robust RL. In the absence of normalization, noise of the same scale will have a minimal impact on dimensions with large scale values, but a significant impact on dimensions with small scale values. The corruption is biased for different dimensions. Although this issue may not be crucial considering the overall contribution of this work, I believe it can be addressed by (1) emphasizing the unnormalized setting in the main paper and mentioning it in the limitation section, and (2) providing some results in the normalized setting if possible.
>
> We noticed that some works applied the perturbations on normalized states [1,2] while some did not [3,4]. We acknowledge that applying normalization on state observations will balance the corruption effects on different state dimensions. Following the reviewer's suggestion, we added discussions to emphasize the unnormalized setting in the main paper (Section 4) as follows:
>
> **We remark that we do not normalize the
> state observations when applying the perturbations as in Shen et al. (2020); Sun et al. (2021), in
> contrast to Zhang et al. (2020); Yang et al. (2022).**
>
> Further, we have added a new paragraph to discuss this issue in Appendix E. Limitation:
>
> **Another limitation of this work is the lack of validation of DMBP’s performance with normalized
> state observations (as in Zhang et al. (2020); Yang et al. (2022)). Without normalizing the state
> observation, adding noise of the same scale may introduce a bias in the corruption’s effect on different
> dimensions. In future work, we will validate and optimize the performance of DMBP with normalized
> state observations.**
>
> Unfortunately, we struggle to provide a comparison in the normalized setting during the rebuttal period due to time constraints. We will add this point in our future work.
>
> > Another concern I have is the considerable computational cost of DMBP compared to other algorithms. I recommend that the authors include a discussion on potential future strategies to mitigate this issue and emphasize it as a limitation in the corresponding section.
>
> As we introduced Appendix D.5, DMBP demonstrates faster convergence compared to baseline algorithms, albeit at a higher computational cost. To make it clear, we have emphasized this point in the corresponding section:
>
> **Although the training of DMBP requires more computing time per epoch,
> DMBP expedites the convergence (in 200-300 epochs) of the baseline algorithms (which usually converge in 1000-3000 epochs),
> i.e., the total training time of DMBP is similar to that of the baseline algorithms in comparison.**
>
> [1] Huan Zhang, Hongge Chen, Chaowei Xiao, Bo Li, Mingyan Liu, Duane Boning, and Cho-Jui
> Hsieh. Robust deep reinforcement learning against adversarial perturbations on state observations.
> Advances in Neural Information Processing Systems, 33:21024–21037, 2020.
>
> [2] Rui Yang, Chenjia Bai, Xiaoteng Ma, Zhaoran Wang, Chongjie Zhang, and Lei Han. RORL: Robust
> offline reinforcement learning via conservative smoothing. In Advances in Neural Information
> Processing Systems, 2022.
>
> [3] Qianli Shen, Yan Li, Haoming Jiang, Zhaoran Wang, and Tuo Zhao. Deep reinforcement learning
> with robust and smooth policy. In International Conference on Machine Learning, pp. 8707–8718.
> PMLR, 2020.
>
> [4] Yanchao Sun, Ruijie Zheng, Yongyuan Liang, and Furong Huang. Who is the strongest enemy?
> Towards optimal and efficient evasion attacks in deep RL. arXiv preprint arXiv:2106.05087, 2021.

---

> > ### Comment · Reviewer_As3y · 2023-11-22
> > **Thanks for your response**
> >
> > Thank you for your prompt response. I appreciate that my first concern has been addressed, but the second concern regarding potential future strategies to expedite training and evaluation still remains. I hope the authors can engage in an insightful discussion on this matter in the revision. I will maintain my current rating.

---

> ### Author Response · Authors · 2023-11-22
> **Thanks for your suggestions**
>
> Thank you for the further suggestions. We have added a paragraph in Appendix D.5 as follows:
>
> **To expedite the training process of DMBP, one can optimize the training schedule by adjusting the number of diffusion steps (Chung et al., 2022; Franzese et al., 2022) and the design of noise scale (Nichol & Dhariwal, 2021) through learning, which have been proven effective in speeding up the convergence rate. Analogously, to expedite the testing process of DMBP, one can employ implicit sampling techniques, such as Song et al. (2020a); Zhang et al. (2022a), which aim to reduce the number of reverse denoising steps required during the testing phase. The proposed methods will be subject to verification in our future work.**
>
> [1] Hyungjin Chung, Byeongsu Sim, and Jong Chul Ye. Come-closer-diffuse-faster: Accelerating conditional diffusion models for inverse problems through stochastic contraction. In Proceedings of the IEEE/CVF Conference on Computer Vision and Pattern Recognition, pp. 12413–12422, 2022.
>
> [2] Giulio Franzese, Simone Rossi, Lixuan Yang, Alessandro Finamore, Dario Rossi, Maurizio Filippone, and Pietro Michiardi. How much is enough? a study on diffusion times in score-based generative models. arXiv preprint arXiv:2206.05173, 2022.
>
> [3] Alexander Quinn Nichol and Prafulla Dhariwal. Improved denoising diffusion probabilistic models. In International Conference on Machine Learning, pp. 8162–8171. PMLR, 2021.
>
> [4] Jiaming Song, Chenlin Meng, and Stefano Ermon. Denoising diffusion implicit models. arXiv preprint arXiv:2010.02502, 2020a.
>
> [5] Qinsheng Zhang, Molei Tao, and Yongxin Chen. gddim: Generalized denoising diffusion implicit models. arXiv preprint arXiv:2206.05564, 2022a.

---

> > ### Comment · Reviewer_As3y · 2023-11-23
> > **Thanks for your response**
> >
> > Thanks for your response. I have no additional concerns and  I will maintain my positive rating about this work.

---

### Official Review · Reviewer_FLnj · 2023-10-31

**Soundness:** 3 good
**Presentation:** 2 fair
**Contribution:** 3 good
**Rating:** 6
**Confidence:** 4

**Summary:**

Offline reinforcement learning (RL) enables training without real-world interactions, but faces challenges with robustness against state observation perturbations caused by factors like sensor errors and adversarial attacks. The Diffusion Model-Based Predictor (DMBP) framework has been introduced to address these issues by predicting actual states using conditional diffusion models, focusing on state-based RL tasks. Unlike traditional methods, DMBP leverages diffusion models as noise reduction tools, enhancing the resilience of existing offline RL methods against various state observation perturbations. The framework utilizes a conditioned diffusion model to estimate current states by reversely denoising data and incorporates a non-Markovian loss function to prevent error accumulation. DMBP's advantages include improved robustness against different scales of noise and adversarial attacks, as well as the ability to manage incomplete state observations, making it suitable for real-world scenarios like robots operating with malfunctioning sensors.

**Strengths:**

1. DMBP strengthens the resilience of existing offline RL algorithms, allowing them to handle different scales of random noises and adversarial attacks effectively.
2. Unlike traditional approaches, DMBP leverages diffusion models primarily for noise reduction, rather than as generation models. This innovative use helps in better state prediction and recovery against observation perturbations.
3. The framework introduces a non-Markovian loss function, specifically designed to prevent the accumulation of estimation errors over the RL trajectory.
4. DMBP offers a more balanced approach than methods that train robust policies against worst-case disturbances. This ensures that policies do not become overly conservative, which can hinder performance in certain scenarios.
5. DMBP's inherent ability, derived from the properties of diffusion models, allows it to effectively manage situations with incomplete state observations. This is particularly valuable in real-world applications, such as when robots operate with compromised sensors.

**Weaknesses:**

1. The paper has the assumption that perturbation on state space follows a Gaussian distribution. However, in practical settings, such perturbations might be biased and skewed. For example, a water drop on a camera might lead to distortion of captured images. Therefore, it might be better if the authors could elaborate on the assumption of the perturbations and how they are produced.
2. Introducing a diffusion model for noise reduction could increase the complexity of the model, which may lead to longer training time and resource-intensive computations. The authors might want to include more details regarding training time and computational resources.
3. Scaling to very large state spaces or handling very noisy environments might pose challenges, especially when using diffusion models. It would be better for the authors to consider testing on tasks with larger state spaces such as the humanoid.
4. Eq.(2) is wrong as the recovered state should not be a probability.

**Questions:**

Same as the weakness.

---

> ### Author Response · Authors · 2023-11-15
> **Response**
>
> We acknowledge Reviewer FLnj for providing the helpful suggestions. Please see our response to your concerns below.
>
> 1. We acknowledge that the assumption of Gaussian distributed noise has certain limitations. It is commonly used as a standard setting for benchmark in many related research works [1, 2, 3]. Moreover, our experiments have incorporated different types of non-Gaussian distributed noises, including uniformly distributed random noise, maximum action-difference attack, and minimum Q-value attack. Under these adversarial noise distributions that have been commonly adopted in the literature, the proposed DMBP, along with the baseline algorithms, can achieve satisfactory denoising performance
> (cf. Tables 2, 7, 8, 9, and Fig. 9).
>
> 2. In the revised manuscript, we have added a new section (Appendix D.5) on the computing costs associated with DMBP (including training and testing costs).
>
> 3. We agree with the reviewer's point that larger state spaces may pose challenges for DMBP in accurately recovering actual states from noisy ones. Further investigation is required to confirm this. However, it is noted that most offline RL algorithms struggle to learn satisfactory policies when the state dimension is extremely large, such as in the humanoid environment with a total state dimension of 376.
> As a benchmark that is commonly used in most
> offline DRL works, D4RL does not provide datasets for the humanoid environment. To our knowledge, the environment with the highest state space in the D4RL benchmark is Franka Kitchen, with a state dimension of 59. The experimental results in Table 9 and Fig. 14 demonstrate that the proposed DMBP performs well in this high-dimensional environment.
>
> 4. We apologize for the mistake we made. In the revised manuscript, it has been fixed as follows:
>
> The denoised state $\hat{{s}}\_{t}$ is sampled from the reverse denoising process, which can be expressed as a Markov chain:
>     $\hat{{s}}\_{t} \sim p\_{\theta}(\tilde{{s}}^{0:k}\_{t} \mid {a}\_{t-1}, \tau^{\hat{{s}}}\_{t-1}) = f\_k(\tilde{{s}}\_{t}) \prod\limits\_{i=1}^{k} p\_{\theta}(\tilde{{s}}^{i-1}\_{t} \mid \tilde{{s}}^{i}\_{t}, {a}\_{t-1}, \tau^{\hat{{s}}}\_{t-1}),$
>
> [1] Matteo Turchetta, Andreas Krause, and Sebastian Trimpe. Robust model-free reinforcement learning
> with multi-objective bayesian optimization. In 2020 IEEE International Conference on Robotics
> and Automation (ICRA), pp. 10702–10708. IEEE, 2020
>
> [2] Aounon Kumar, Alexander Levine, and Soheil Feizi. Policy smoothing for provably robust reinforce-
> ment learning. arXiv preprint arXiv:2106.11420, 2021.
>
> [3] Ke Sun, Yingnan Zhao, Shangling Jui, and Linglong Kong. Exploring the training robustness
> of distributional reinforcement learning against noisy state observations. In Joint European
> Conference on Machine Learning and Knowledge Discovery in Databases, pp. 36–51. Springer,
> 2023.

---

> > ### Comment · Reviewer_FLnj · 2023-11-22
> > **Thank you for the authors' response.**
> >
> > Thank you for the response which has resolved most of my concerns. I will raise my score.

---

### Official Review · Reviewer_YkkL · 2023-10-31

**Soundness:** 3 good
**Presentation:** 3 good
**Contribution:** 3 good
**Rating:** 6
**Confidence:** 4

**Summary:**

This paper addresses the challenges in offline reinforcement learning (RL), particularly the robustness against state perturbations. The authors introduce the Diffusion Model-Based Predictor (DMBP), a novel framework that employs conditional diffusion models to recover actual states in state-based RL tasks and proposes a non-markovian loss function to mitigate error accumulation. The framework is designed to handle incomplete state observations and various scales of random noises and adversarial attacks, improving the robustness of existing offline RL algorithms. DMBP is empirically evaluated, demonstrating significant performance improvements in terms of robustness against different perturbations on state observations without leading to over-conservative policies.

**Strengths:**

1. It is significant for this paper to promote the robustness of offline RL methods.
2. This paper shows its novelty by applying diffusion method to improve the robustness of offline RL against state perturbations, especially designing the non-Markovian loss function to reduce the accumulated error in state estimation.
3. The proposed method demonstrates strong performance and has been extensively evaluated with different offline RL methods and noise types and in various benchmark tasks.
4. This paper is generally well written and easy to follow and covers well related works.

**Weaknesses:**

1. One main concern is that observation perturbations can be potentially addressed by traditional methods, especially when we assume observation perturbations does not affect reward and transition functions. The reviewer is curious of how offline RL methods with Kalman Filter perform compared to DMBP?
2. Minor issue: all results in tables should be shown with standard deviations.

**Questions:**

1. In the experiments, results show DMBP significantly improve the robustness performance of existing offline RL methods. The review noticed that most offline RL methods used are value-based. How will DMBP perform when used with weighted imitation learning methods, e.g., IQL?
2. Is DMBP sensitive to hyperparameters (i.e., a, b, c) of the variance schedule? Any guidance to choose these hyperparameters?
3. It is intutitve that diffusion denoising can perform well for random noise. Can the authors provide insights why it also works well for adversarial attacks?

---

> ### Author Response · Authors · 2023-11-15
> **Response (Part1/2)**
>
> We thank Reviewer YkkL for the compliments on the novelty of our research and valuable suggestions.
> We address the concerns in your review point by point below.
>
> Response to your Questions
>
> 1. We agree that imitation learning is a class of highly effective and well-studied offline RL algorithms. Adding an imitation learning algorithm to the baseline algorithms would be valuable to demonstrate the generality of DMBP. However, given the space constraints and limited time, we struggled to do it for now. It should be noted that a class of value-based offline RL algorithms, which utilize policy regularization methods, achieve a similar effect as imitation learning, i.e., selectively cloning the policy in datasets. The baseline algorithms adopted in our experiments, like BCQ, TD3+BC, and Diffusion QL, belong to this category.  Experimental results demonstrate that DMBP performs well for baseline algorithms in this category.
> In the revised manuscript, we have added a new section (Appendix D.6) to discuss the observed experimental results, where we analyze the performance of DMBP in comparison to policy regularization category and conservative Q estimation category offline RL algorithms.
> Below we present partial experimental results on the suggested algorithm IQL (Implicit Q Learning), which
> is trained on expert dataset in Mujoco domain.
>
>  Noise scale | Noise type | HalfCheetah|                 |Hopper      |                    |Walker2d |                  |
> | ------------  |------------  | ------------   | ------------| ------------| ------------| ------------| ------------ |
> |                    |                   | Base           | DMBP       | Base        | DMBP       | Base        | DMBP        |
> |      0      |        -     | 84.7$\pm$14.6  |             | 103.4$\pm$21.5 |           | 109.3$\pm$0.9  |             |
> |      0.05      |        G-rand     | 15.2$\pm$9.9  | 53.0$\pm$17.6 | 28.5$\pm$27.4 | 58.7$\pm$15.1 | 105.5$\pm$8.4 | 105.6$\pm$11.6 |
> |           |  U-rand     | 21.1$\pm$13.0 | 61.6$\pm$18.6 | 43.3$\pm$34.3 | 69.2$\pm$28.4 | 108.5$\pm$1.3 | 109.4$\pm$1.0  |
> |               | MAD | 14.9$\pm$11.0 | 37.5$\pm$14.2 | 19.8$\pm$16.8 | 38.3$\pm$14.4 | 104.3$\pm$10.1 | 101.4$\pm$20.4 |
> |           | MinQ | 25.5$\pm$15.8 | 46.8$\pm$14.2 | 26.0$\pm$24.6 | 31.6$\pm$18.8 | 106.5$\pm$1.5 | 103.4$\pm$11.4 |
> |  0.10  | G-rand | 9.3$\pm$6.1 | 20.6$\pm$11.8 | 11.5$\pm$6.6 | 24.4$\pm$11.1 | 58.5$\pm$23.7 | 83.9$\pm$21.8 |
> |          |  U-rand |  12.8$\pm$9.4 |  29.8$\pm$17.0 |  26.6$\pm$22.5 |  32.6$\pm$19.1 |  79.9$\pm$17.4 |  103.5$\pm$6.5 |
> |         |  MAD |  7.7$\pm$5.3 |  17.8$\pm$9.8 |  10.7$\pm$8.2 |  17.7$\pm$8.5 |  42.2$\pm$30.4 |  72.5$\pm$19.6 |
> |         |  MinQ |  10.0$\pm$7.1 |  26.6$\pm$14.3 |  8.5$\pm$8.6 |  15.6$\pm$9.5 |  77.5$\pm$35.7 |  98.6$\pm$8.8 |
>
> 2. There is no need to individually select the hyperparameters related to the variance schedule (i.e., $a, b, c$), and one can use the same set of predetermined values (as listed in Table 3) for all environments and datasets.
> We redesigned our variance schedule with the specific aim of effectively handling information with small to medium scale noises. In other words, we intentionally set $\sqrt{\bar{\alpha}\_i}$  to be large and $\sqrt{1 - \bar{\alpha}\_{i}}$ to be small for smaller diffusion timesteps $i$, which ensures that ${s}\_{t}^i=\sqrt{\bar{\alpha}\_i} {s}\_{t} + \sqrt{1-\bar{\alpha}\_i} {\epsilon}$ maintains significant raw state information.
>
> 3. In classical diffusion models, one typically assumes that the noise follows a Gaussian distribution and uses batch-generated Gaussian distributed noise for training. However, during the testing phase, the noised states are generated individually. In other words, for a single instance of $\tilde{{s}}_t = {s}_t + {\epsilon}_t$, we cannot determine the exact distribution of ${\epsilon}_t$ based on a single sample.
> For DMBP, during the single-step denoising process in testing, when the noise scale is given, it is trained to reconstruct ${s}_t$ from the observed $\tilde{{s}}_t$ for all possible values of the noise ${\epsilon}_t$ within a given range, which means that DMBP aims to learn a mapping that can effectively denoise the observed state within a certain range, regardless of the specific distribution of the noise. This may explain why DMBP also works well for adversarial attacks.

---

> ### Author Response · Authors · 2023-11-15
> **Response (Part2/2)**
>
> Response to the listed Weaknesses
>
> 1. We do agree with your point that traditional methods (e.g., the Kalman Filter for discrete control) have been proposed to address the issue of observation perturbations for discrete scenarios. However, the Kalman filter requires exact knowledge of the governing equation of the controlled system and the related parameters.
> In offline RL settings, on the other hand, we have no prior knowledge on the system's governing equations or parameters (otherwise, the agent can be trained online with a simulator). The only information available is the offline dataset. Modeling the environment for high action/state space problems with limited data can be prohibitively challenging. As discussed in the Introduction, modeling environment (called model-based) methods are mostly used for data augmentation in offline RL. To our knowledge, there are no relevant references on offline RL methods using the Kalman Filter for continuous control problems.
>
> 2. Thanks for pointing this out! We have fixed the issue and supplemented the data in all tables with standard deviations.

---

> > ### Comment · Reviewer_YkkL · 2023-11-23
> >
> > I appreciate the detailed response. I will maintain my rating.

---

### Official Review · Reviewer_ad2c · 2023-11-01

**Soundness:** 3 good
**Presentation:** 4 excellent
**Contribution:** 3 good
**Rating:** 8
**Confidence:** 3

**Summary:**

When facing observation perturbations, reinforcement learning agent may perform very poorly. Unlike many prior work which train a robust policy, this paper instead utilizes diffusion model as noise reduction tool to retrieve accurate state information. Then the proposed framework feeds the de-noised state to any offline RL algorithm. Additional techniques are introduced, e.g., specialized loss to facilitate multi-step diffusion accuracy. Extensive simulations are conducted on MuJoCo control tasks.

**Strengths:**

1. The proposed framework uses the latest diffusion model to solve offline RL with state perturbations. I believe there is a strong motivation behind this approach. Indeed, an important RL problem is studied with state-of-the-art tool.

2. The paper is well-written. The motivations behind each section is quite clear. I enjoyed reading the paper a lot.

3. I believe the contribution is solid in this paper: (a) proposed a non-Markovian loss that facilitates multi-step diffusion accuracy; (b) a tractable version of its VLB is proposed; (c) thorough simulation and ablation studies

4. The proposed framework is very flexible and can work with any good offline RL algorithm.

**Weaknesses:**

There is no notable weakness in my opinion.

**Questions:**

1. When training different algorithms, do you use the same dataset size and batch size for all algorithms?

2. In offline RL, especially the theoretical community, we are often concerned with the data coverage problem (full coverage vs partial coverage). I am interested in the performance of DMBP in this scenario. In particular, D4RL has millions of transitions in its dataset. Although one can argue that this is not that much for a continuous control task, we sometimes see more extreme case in tabular setting where the data is extremely limited, e.g., barely supporting the optimal policy's state-action visitation. I conjecture that if DMBP is paired with algorithm like VI-LCB[1] which is proven to be statistically efficient, it can also facilitate offline RL with scarce data setting. Do you have any insight on this?

[1]Li, G., Shi, L., Chen, Y., Chi, Y., and Wei, Y. (2022a). Settling the sample complexity of model-based
offline reinforcement learning. arXiv preprint arXiv:2204.05275

Overall I think this is a great paper, and it should be a good contribution to ICLR.

---

> ### Author Response · Authors · 2023-11-15
> **Response**
>
> We thank Reviewer ad2c for providing the positive feedback and recognizing the importance of our work. Please see our response to your questions below.
>
> 1. When we train different baseline algorithms (including BCQ, CQL, TD3+BC, Diffusion QL, and RORL), we use the same dataset size and batch size.
> By utilizing the datasets in the D4RL benchmark, we ensure that the dataset size remains fixed, thereby ensuring fairness in the comparisons made.
> Furthermore, we adhere to the suggested hyperparameter settings for all baseline algorithms, including setting the batch size to 256.
>
> 2. We agree that the data convergence issue of a RL algorithm significantly affects its stability and performance. In the D4RL benchmark, datasets  labelled with "medium-replay" in the Mujoco domain have only 1/5 of the data compared to other datasets. In addition, a significant portion of the state-action visitations are from suboptimal strategies, and many are from random policy. In our experiments, it is found that DMBP still performs well under such circumstances (cf. Tables 1 and 2). The baseline algorithm VI-LCB recommended by the reviewer has strong theoretical guarantees. However, due to the time constraints, we are currently unable to present the simulation results of DMBP assisting VI-LCB in a scarce data setting with perturbed state observations. We will validate this in our future work.

---

> > ### Comment · Reviewer_ad2c · 2023-11-21
> >
> > I thank authors for the feedback. I will maintain my rating.

---

### Author Response · Authors · 2023-11-15
**Response to all**

We would like to express our sincere gratitude to the reviewers for their dedicated time in reviewing our work. We truly appreciate the professional and constructive suggestions and are encouraged by the recognition of the novelty and significance of our work. Below we provide responses and clarifications for each reviewer's suggestions and comments.

We have also revised our manuscript to address all reviewers' concerns. All changes in the revised manuscript are marked in blue. A summary of the revisions is listed below.

1. We added a paragraph to review the training-time and testing-time robustness of RL in Section 2.

2. We supplemented the data in all tables with standard deviation.

3. We added a new section (Appendix D.5) on the computing costs (including training costs and testing costs).

4. We added a new section (Appendix D.6) to discuss the robustness enhancement achieved by DMBP. We also utilized t-SNE to visualize the state and action distributions, providing insights into the underlying reasons behind the observed performance.

5. We added some clarification (in Section 4 and Appendix E) on our choice not to normalize the datasets.

6. We fixed some typos and an incorrectly expressed equation.

---

### Author Response · Authors · 2023-11-21
**Looking forward to Reviewers' feedbacks**

Dear Reviewers:

Many thanks for your dedicated review and constructive suggestions, which have led to significant improvements in the revised manuscript. As the deadline for revision and discussion is approaching (on November 22, at 23:59 UTC-12h), we would like to follow up and seek your further valuable feedback. We would greatly appreciate it if you could re-evaluate our manuscript and provide additional insightful comments. We sincerely hope to continue the discussion with you and address any remaining concerns.

Best regards,

Authors of Paper Submission #2608

---

### Meta-Review · Area_Chair_tHkP · 2023-12-05

**Metareview:**

This paper addresses the robustness of offline RL with respect to state observation perturbations.  The proposed solution uses Diffusion Model-Based Predictor (DMBP) to predict the states, improving the resilience by reducing the noise.  A non-Markovian loss function is also introduced to prevent error accumulation.

The paper is well written.  The idea is interesting, and the experimental results show it is effective.  All reviewers and myself find the paper a good addition to the conference.

**Justification For Why Not Higher Score:**

The scalability of the method can be better addressed.

**Justification For Why Not Lower Score:**

The paper is well written.  The idea is interesting, and the experimental results show it is effective.  All reviewers and myself find the paper a good addition to the conference.

---

### Decision · Program_Chairs · 2024-01-16

Accept (poster)